# MODEL-FREE CONTROL OF NONLINEAR STOCHASTIC SYSTEMS WITH STABILITY GUARANTEE

## ABSTRACT

Reinforcement learning (RL) offers a principled way to achieve the optimal cumulative performance index in discrete-time nonlinear stochastic systems, which are modeled as Markov decision processes. Its integration with deep learning techniques has promoted the field of deep RL with an impressive performance in complicated continuous control tasks. However, from a control-theoretic perspective, the first and most important property of a system to be guaranteed is stability. Unfortunately, stability is rarely assured in RL and remains an open question. In this paper, we propose a stability guaranteed RL framework which simultaneously learns a Lyapunov function along with the controller or policy, both of which are parameterized by deep neural networks, by borrowing the concept of Lyapunov function from control theory. Our framework can not only offer comparable or superior control performance over the state-of-the-art RL algorithms, but also construct a Lyapunov function to validate the closed-loop stability. In the simulated experiments, our approach is evaluated on several well-known examples including classic CartPole balancing, 3-dimensional robot control and control of synthetic biology gene regulatory networks. Compared with RL algorithms without stability guarantee, our approach can enable the system to recover to the operating point when interfered by uncertainties such as unseen disturbances and system parametric variations to a certain extent. (Anonymous code is available to reproduce the experimental results[1].)

## 1 INTRODUCTION

Control of discrete-time nonlinear stochastic systems is an important topic in both control theory and reinforcement learning. In the past decades, the advancement of nonlinear control theory in the control community has been successfully applied in aircraft, automobiles, advanced robots and space systems (Slotine et al., 1991; Isidori, 1995). Concurrently, reinforcement learning was developed in the machine learning community to address similar nonlinear control problems (Sutton et al., 1992; Tesauro, 1995; Bertsekas & Tsitsiklis, 1996). Until recently, significant progress has been made by combining advances in deep learning (LeCun et al., 2015) with reinforcement learning. Impressive results are obtained in a series of high-dimensional continuous nonlinear control problems (Duan et al., 2016; Zhang et al., 2016; Zhu et al., 2017; Gu et al., 2017) in which control-theoretic approach is typically difficult to apply.

Given a control system, regardless of which controller design method is used, control theory or reinforcement learning, the first and most important property of a system needs to be guaranteed is stability, because an unstable control system is typically useless and potentially dangerous (Slotine et al., 1991). Qualitatively, a system is described as stable if starting the system in the neighborhood of its desired operating point implies that it will stay around the point ever after. For aircraft control systems, a typical stability problem is intuitively related to the following question: will a trajectory perturbation caused by a gust result in a significant deviation in the later flight trajectory? Here, the desired operating point of the system is the flight trajectory in the absence of disturbance. Every control system, whether linear or nonlinear, involves a stability problem which should be carefully studied.

---

[1] https://www.dropbox.com/sh/j9mhvi0vydu7x7c/AACwJbqU5MCLcKPGgcOv0zrHa?dl=0

The most useful and general approach for studying the stability of control systems is Lyapunov method (Lyapunov, 1892), which is dominant in control engineering (Åström & Wittenmark, 1989; Mayne et al., 2000). In Lyapunov method, a scalar "energy-like" function called Lyapunov function is constructed to analyze the stability of the system. For a linear dynamical system, a quadratic function is typically chosen as Lyapunov function in some classic controller design method, such as linear quadratic regulator (LQR) and model predictive control (MPC). Unfortunately, there is no universal method for constructing Lyapunov functions.

In this paper, we propose a stability guaranteed reinforcement learning framework to jointly learn the controller or policy[2] and a Lyapunov function both of which are parameterized by deep neural networks, with a focus on stabilization and tracking problems in discrete-time nonlinear stochastic systems modeled by Markov decision process. The contribution of our paper can be summarized as follows: 1) a novel data-based approach for analyzing the stability of the closed-loop system is proposed by constructing a Lyapunov function parameterized by deep neural network; 2) a practical learning algorithm is designed to search the stability guaranteed controller; 3) the learned controller is able to stabilize the system when interfered by uncertainties such as unseen disturbance and system parameters variations of certain extent. In our experiment, we can show that the stability guaranteed controller is more capable of handling uncertainties compared to those without such guarantees in nonlinear control problems including classic CartPole stabilization tasks, control of 3D legged robots and manipulator and reference tracking tasks for synthetic biology gene regulatory networks.

## 1.1 RELATED WORKS

In model-free reinforcement learning (RL), stability is rarely addressed due to the formidable challenge of analyzing and designing the closed-loop system dynamics in a model-free manner (Buşoniu et al., 2018), and the associated stability theory in model-free RL remains as an open problem (Buşoniu et al., 2018; Gorges, 2017).

Recently, Lyapunov analysis is used in model-free RL to solve control problems with safety constraints (Chow et al., 2018; 2019). In Chow et al. (2018), Lyapunov-based approach for solving constrained Markov decision process is proposed with a novel way of constructing the Lyapunov function through linear programming. In Chow et al. (2019), the above results were further generalized to continuous control tasks. Even though Lyapunov-based methods were adopted in these results, neither of them addressed the stability of the system.

As a basic tool in control theory, the construction/learning of Lyapunov function is not a trivial issue and many works are devoted to this problem. In Perkins & Barto (2002), the RL agent controls the switch between controllers designed using Lyapunov domain knowledge, so that any policy is safe and reliable. Petridis & Petridis (2006) proposes a straightforward approach for constructing Lyapunov function for nonlinear systems using neural networks. Richards et al. (2018) proposes a learning-based approach for constructing Lyapunov neural networks with maximized region of attraction. Results on learning and construction of Lyapunov functions are referred to Noroozi et al. (2008); Prokhorov (1994); Serpen (2005); Prokhorov & Feldkamp (1999).

Other interesting results on the stability of learning-based control systems are reported in recent years. In Postoyan et al. (2017), an initial result is proposed for the stability analysis of deterministic nonlinear systems with optimal controller for infinite-horizon discounted cost, based on the assumption that discount is sufficiently close to 1. In Berkenkamp et al. (2017), a learning model-based safe RL approach with safety guarantee during exploration is introduced but limited to Lipschitz continuous nonlinear systems such as Gaussian process model. In addition, the verification of stability condition requires the discretization of state space, which limits its application to tasks with low-dimensional finite state space.

## 2 PROBLEM STATEMENT

We consider discrete-time nonlinear stochastic systems modeled by the Markov decision process (MDP). A MDP is defined by the tuple $(\mathcal{S}, \mathcal{A}, c, P, \rho)$, where $\mathcal{S} \subseteq \mathbb{R}^n$ is the set of states, $\mathcal{A} \subseteq \mathbb{R}^m$ is the set of actions, $c(s, a) : \mathcal{S} \times \mathcal{A} \to \mathbb{R}_+$ is the cost function, $P(s'|s, a)$ is the transition probability function, and $\rho(s)$ is the starting state distribution.

---

[2]Controller and policy will be used interchangeably throughout the paper.

In this paper, we focus on the stabilization and tracking problems for discrete-time nonlinear stochastic systems modeled by MDP. For both problems, the goal is to find a policy $\pi$ which can bring the cost $c$ to zero. In stabilization problems, the cost function is defined as the norm of states $\|s\|$ where $\|\cdot\|$ denotes the Euclidean norm. In tracking problems, we divide the state $s$ into two vectors, $s^1$ and $s^2$, where $s^1$ is composed of elements of $s$ that are aimed at tracking the reference signal $r$ while $s^2$ contains the rest. For tracking, $\|s^1 - r\|$ is chosen to be the cost function.

From a control theoretic perspective, the task of stabilization and tracking could be addressed as ensuring the closed-loop system or error system to be asymptotically stable, i.e., starting from an initial point, the trajectories of state always converge to a single point or the reference trajectory. Let $c_\pi(s) \triangleq \mathbb{E}_{a\sim\pi} c(s,a)$ denote the cost function under policy $\pi$, the definition of stability studied in this paper is given as follows.

**Definition 1** *The stochastic system is said to be stable in mean cost if $\lim_{t\to\infty} \mathbb{E}_{s_t} c_\pi(s_t) = 0$ holds for any initial condition $s_0 \in \{s_0 | c_\pi(s_0) \leq b\}$. If $b$ is arbitrarily large then the stochastic system is globally stable in mean cost.*

**Remark 1** *Form of the cost is strictly ruled as the Euclidean norm of state or partial state, while other forms are not considered in this paper. The stability studied in this paper is a type of local asymptotic stochastic stability, which is different to the definition of mean square stability (MSS) that extensively studied on stochastic systems in control theory (Shaikhet, 1997; Huang, 2012).*

Before proceeding, some notations are to be defined. The closed-loop transition probability is denoted as $P_\pi(s'|s) \triangleq \int_A \pi(a|s) P(s'|s,a) \mathrm{d}a$. We also introduce the closed-loop state distribution at certain instant $t$ as $P(s|\rho,\pi,t)$, which could be defined in an iterative way: $P(s'|\rho,\pi,t+1) = \int_S P_\pi(s'|s) P(s|\rho,\pi,t) \mathrm{d}s, \forall t \in \mathbb{Z}_+$ and $P(s|\rho,\pi,0) = \rho(s)$.

## 3 MAIN RESULTS

In this section, we propose the main assumptions and a new theorem.

**Assumption 1** *The stationary distribution of state $q_\pi(s) \triangleq \lim_{t\to\infty} P(s|\rho,\pi,t)$ exists.*

**Assumption 2** *There exists a positive constant $b$ such that $\rho(s) > 0, \forall s \in \{s | c_\pi(s) \leq b\}$.*

Our approach is to construct/find a Lyapunov function which can be used to analyze the stability of the closed-loop system. The Lyapunov method has long been used for stability analysis and controller design in control theory (Boukas & Liu, 2000), but mostly exploited along with a *known* model, whether deterministic or probabilistic (Corless & Leitmann, 1981; Thowsen, 1983; Huang et al., 2011).

The Lyapunov function is a class of semi-positive definite functions $L : \mathcal{S} \to \mathbb{R}_+$. The general idea of exploiting Lyapunov function is to ensure that the difference (or derivative, if the system is in continuous time) of Lyapunov function along the state trajectory is semi-negative definite, so that the state goes in the direction of decreasing the value of Lyapunov function and eventually converges to the origin or a sub-level set of Lyapunov function. Next, we give sufficient conditions for a system to be stable in mean cost in the following.

**Theorem 1** *The stochastic system is stable in mean cost if there exists a function $L : \mathcal{S} \to \mathbb{R}_+$ and positive constants $\alpha_1$, $\alpha_2$ and $\alpha_3$, such that*

$$\alpha_1 c_\pi(s) \leq L(s) \leq \alpha_2 c_\pi(s) \tag{1}$$

$$\mathbb{E}_{s\sim\mu_\pi}(\mathbb{E}_{s'\sim P_\pi} L(s') - L(s)) \leq -\alpha_3 \mathbb{E}_{s\sim\mu_\pi} c_\pi(s) \tag{2}$$

*where $\mu_\pi(s) \triangleq \lim_{N\to\infty} \frac{1}{N} \sum_{t=0}^{N} P(s_t = s|\rho,\pi,t)$ is the sampling distribution.*

Due to space limitations, we will include the detailed proof in Appendix A. Eq. (2) is called the energy decreasing condition, i.e., requiring the expectation of Lyapunov function to be decreasing between two consecutive instants. Eq. (1) is the constraint for Lyapunov function, though a rather broad range of parameterization is covered. The sum of quadratic polynomials, e.g., $L(s) = s^T Q s$ where $Q$ is a

positive definite matrix, are extensively used in the control theory. Such Lyapunov functions can be efficiently discovered by the semi-definite programming solvers and bring in limited conservatism for the control tasks where the cost are also of a quadratic form. In (Richards et al., 2018), a neural network $\phi_\theta(\cdot)$ is designed to construct the Lyapunov function, $L(s) = \phi_\theta(s)^T \phi_\theta(s)$. As explored in (Chow et al., 2018) and (Berkenkamp et al., 2017), the value function could be exploited as a Lyapunov function as well. Additionally, the sum of cost over a limited time horizon could also be employed as Lyapunov function, i.e., $L(s) = \sum_t^{t+N} \mathbb{E} c_\pi(s_t)$, which is a valid Lyapunov candidate in model predictive control literature (Mayne & Michalska, 1990; Mayne et al., 2000).

The choice of Lyapunov function candidate plays an important role in learning a policy. Value function evaluates the infinite time horizon and thus offers a better performance in general, but is rather difficult to approximate because of significant variance and bias (Schulman et al., 2015). On the other hand, the finite horizon sum of cost provides an explicit target for learning a Lyapunov function, thus inherently reduces the bias and enhances the learning process. However, as the model is unknown, predicting the future costs based on the current state and action inevitably introduces variance, which grows as the prediction horizon extends. In principle, for tasks with simple dynamics, the sum-of-cost choice enhances the convergence of learning and robustness of the trained policies, while for complicated systems the choice of value function generally produces better performance. In this paper, we use both value function and sum-of-cost over various horizons as Lyapunov function candidates in different tasks and compare their strength and weakness respectively. Now we would like to give the following two remarks.

**Remark 2** *This remark is on Assumption 1 and sampling distribution $\mu_\pi$. If an MDP is ergodic then the existence of $q_\pi$ is naturally assured, but all states have to be positive recurrent and aperiodic (Papoulis & Pillai, 2002). The existence of sampling distribution $\mu_\pi(s)$ is guaranteed by the existence of $q_\pi(s)$. Since the sequence $\{P(s|\rho, \pi, t), t \in \mathbb{Z}_+\}$ converges to $q_\pi(s)$ as $t$ approaches $\infty$, then by the Abelian theorem, the sequence $\{\frac{1}{N} \sum_{t=0}^N P(s|\rho, \pi, t), N \in \mathbb{Z}_+\}$ also converges and $\mu_\pi(s) = q_\pi(s)$. Thus we use $\mu_\pi$ to approximate the $q_\pi$ since the evaluation of $q_\pi$ requires data to be sampled after infinite instants the episode begins. Even if the sampling period $N << \infty$, one can still assure that $c_\pi$ converges to a neighborhood of zero, which is related to the initial state distribution and length of $N$, i.e., $\mathbb{E}_{s_N} c_\pi(s_N) \leq \frac{\alpha_2}{\alpha_1 + \alpha_3} \mathbb{E}_\rho c_\pi(s_0) - \frac{\alpha_3}{\alpha_1 + \alpha_3} \sum_{t=0}^{N-1} \mathbb{E}_{s_t} c_\pi(s_t)$.*

**Remark 3** *This remark is on the connection to previous results concerning the stability of stochastic systems. It should be noted that the stability conditions of Markov chains have been reported in (Shaikhet, 1997; Meyn & Tweedie, 2012), however, of which the validation requires the full knowledge of the model, i.e., the transition probability $P(s'|s, a)$. On the contrary, our approach solely depends on data to analyze the stability of the closed-loop system, which further enables the model-free learning algorithms with stability guarantee. However, the validation of stability through a sample-based approach theoretically requires tremendous, if not infinite, amount of samples to thoroughly estimate the distributions, which is the drawback of our approach. We would demonstrate empirically that the algorithm built upon this theorem is reliable though only limited sample is collected.*

## 4 ALGORITHM

In this section, we propose an off-policy RL algorithm to learn stability guaranteed policies for discrete-time nonlinear stochastic system modeled by MDP. First, based on the maximum entropy actor-critic framework, we use the Lyapunov function as the critic in the policy gradient formulation. In this algorithm, a Lyapunov critic function $L_c$ is needed, which satisfies $L(s) = \mathbb{E}_{a \sim \pi} L_c(s, a)$. The objective function $J(\pi)$ is given as follows

$$J(\pi) = \mathbb{E}_{(s,a,s',c) \sim \mathcal{D}}[\beta(\log(\pi_\theta(f_\theta(\epsilon, s)|s)) + \mathcal{H}_t) + \lambda(L_c(s', f_\theta(\epsilon, s')) - L_c(s, a) + \alpha_3 c)] \quad (3)$$

where the policy $\pi_\theta$ is parameterized by a deep neural network $f_\theta$, $\epsilon$ is an input vector consisted of Gaussian noise. It should be noted that the Lyapunov critic $L_c(s, a)$ will be parameterized by the square of a neural network to ensure the semi-positive definiteness of Lyapunov function required in Eq.(1), inspired by the structure explored in Richards et al. (2018). More specifically, $L_c(s, a) = \phi^T(s, a)\phi(s, a)$, where $\phi(s, a)$ is a multi-layer fully connected neural network. $\mathcal{D}$ is the distribution of previously sampled states and actions, or a replay buffer. In the above objective, $\beta$

and $\lambda$ are Lagrange multipliers which control the relative importance of policy entropy versus energy decreasing constraint derived from Eq.(2). Similar to Haarnoja et al. (2018), the entropy of policy is expected to remain above the target entropy $\mathcal{H}_t$. The parameters of policy network are updated through gradient descent, where the gradient of Eq.(3) is approximated by

$$\nabla_\theta J(\pi) = \nabla_\theta \beta \log(\pi_\theta(a|s)) + \nabla_a \beta \log(\pi_\theta(a|s))\nabla_\theta f_\theta(\epsilon, s) + \lambda \nabla_{a'} L_c(s', a')\nabla_\theta f_\theta(\epsilon, s') \quad (4)$$

We use $J(L_c)$ in the following equation as the objective function to update the Lyapunov critic,

$$J(L_c) = \mathbb{E}_{\mathcal{D}} \left[\frac{1}{2}(L_c(s, a) - L_{\text{target}}(s, a))^2\right] \quad (5)$$

where $L_{\text{target}}$ is the approximation target for $L_c$.

If the sum of cost is chosen as Lyapunov function candidate, we have

$$L_{\text{target}}(s, a) = \Sigma_t^{t+N} c_t \quad (6)$$

Here, the time horizon $N$ is a hyperparameter to be tuned, of which the influence will be demonstrated in the experiment in Section 5.5. If the value function is chosen as Lyapunov function candidate,

$$L_{\text{target}}(s, a) = c + \gamma L_c'(s', f(\epsilon, s')) \quad (7)$$

where $L_c'$ is the target network parameterized by $\bar{\theta}$ as typically used in the actor-critic methods (Haarnoja et al., 2018; Lillicrap et al., 2015b). $L_c'$ has the same structure with $L_c$, but the parameter is updated through exponentially moving average of weights of $L_c$ controlled by a hyperparameter $\tau$. In fact, the value function is the discounted sum of cost over infinite time horizon. Later in Section 5, we will show the influence of choosing different Lyapunov function candidates.

The value of Lagrange multipliers $\lambda$ and $\beta$ are adjusted by the gradient method maximizing the following two objectives respectively,

$$J(\beta) = \mathbb{E}_{\mathcal{D}} \beta[\log(\pi_\theta(a|s)) + \mathcal{H}_t] \quad (8)$$
$$J(\lambda) = \mathbb{E}_{\mathcal{D}} \lambda[L_c(s', f_\theta(\epsilon, s')) - L_c(s, a) + \alpha_3 c] \quad (9)$$

It should be noted that the value of $\lambda$ and $\beta$ are clipped to be positive. In addition, to prevent $\lambda$ from growing unlimitedly causing the algorithm to diverge, we set an upper bound for $\lambda$. In our experiments, we found that 1 is a suitable value without much further tuning. Pseudo code of the proposed algorithm is shown in Algorithm 1 in Appendix B.

**Remark 4** *The convergence of the algorithm is composed of the convergence of Lyapunov critic $L_c$ and policy $\pi_\theta$ respectively. Empirically, the convergence can be judged by the error of Lyapunov function approximation and value of the Lagrange multiplier (close to zero at convergence). In practice, we found that the algorithm converges well in different experiments without much tuning.*

## 5 EXPERIMENT

In this section, we illustrate four simulated examples to demonstrate the general applicability of the proposed method. First of all, the classic control problem of CartPole balancing from control and RL literature (Barto et al., 1983) is illustrated. Then, we consider more complicated high-dimensional continuous control problem of 3D robots, e.g., HalfCheetah and FetchReach, using MuJoCo physics engine (Todorov et al., 2012). Last, we extend our approach to control robots in nanoscale, i.e., molecular robots. Specifically, we consider the problem of reference tracking for a synthetic biology gene regulatory network known as the Repressilator (Elowitz & Leibler, 2000).

The proposed method is evaluated for the following aspects:

- Convergence: does the proposed training algorithm converge with random parameter initialization and does the stability condition (2) hold for the learned policies;
- Performance: can the goal of the task be achieved or the cumulative cost be minimized;
- Robustness: how do the trained policies perform when faced with uncertainties unseen during training, such as parametric variation and external disturbances;

- Generalization: can the trained policies generalize to follow reference signals that are different from the one seen during training.

We compare our approach with soft actor-critic (SAC) (Haarnoja et al., 2018), one of the state-of-the-art off-policy actor-critic algorithms that outperform a series of off-policy and on-policy methods such as DDPG (Lillicrap et al., 2015a), PPO (Schulman et al., 2017) on the continuous control benchmarks. The variant of safe proximal policy optimization (SPPO) (Chow et al., 2019), a Lyapunov-based method, is also included in the comparison. The original SPPO is developed to deal with constrained MDP, where safety constraints exist. In our experiments, we modify it to apply the Lyapunov constraints on the MDP tasks and see whether it can achieve the same stability guarantee as LAC. In CartPole example, we also compare with linear quadratic regulator (LQR), a classical model-based optimal control method for stabilization.

The outline of this section is as follows. In Section 5.1, a brief introduction will be given on the background and problem description of each example. Then in Section 5.2, the convergence, and performance of the proposed method is demonstrated and compared with SAC. In Section 5.4, the ability of generalization and robustness of the trained policies are evaluated and analyzed. Finally, in Section 5.5, we show the influence of choosing different Lyapunov function candidates upon the performance and robustness of trained policies.

## 5.1 BACKGROUND AND PROBLEM DESCRIPTION

In this section, we will give a brief introduction to the examples considered in this paper. Detailed setup information of the first three examples can be found in Appendix C.

### 5.1.1 CARTPOLE

This is a classical control problem. The controller is to stabilize the pole vertically at a given position. The cost is determined by the norm of the angular position of the pole and the horizontal position of the cart. The control input is the horizontal force $F \in [-20, 20]$ applied in the cart. The agent is dead if the angle $\theta$ between pole and vertical position exceeds a threshold, and the episode ends.

### 5.1.2 HALFCHEETAH

The goal is to control a 17-dimensional 2-legged robot simulated in the MuJoCo simulator. The control task belongs to the reference tracking problem, i.e., to enable the robot to run at the speed of 1m/s in the X-axis direction. The cost is determined by the Euclidean difference between current speed and target speed. The control input is the torque implemented at each joint.

### 5.1.3 FETCHREACH

The agent is to control a simulated manipulator to track a randomly generated goal position with its end effector. The cost is determined by the Euclidean distance between end effector and goal. The control input is the torque implemented at each joint. The manipulator is also simulated in the MuJoCo simulator.

### 5.1.4 REPRESSILATOR

The repressilator is a synthetic biology gene regulatory network with a ring structure pioneered in Elowitz & Leibler (2000), in which each gene represses the other gene cyclically. The dynamics of temporal gene expression exhibit periodic oscillatory behavior. The dynamics of repressilator can be quantitatively described by a set of discrete-time nonlinear difference equations consisting of six states, three mRNAs for transcription and three proteins for translation, based on biochemical kinetic laws. We also include a complicated repressilator example with 4 genes to be controlled, which exhibits an unstable oscillation and is even harder to control.

The objective is to force one protein concentrations to follow a *priori* defined reference trajectories using partially observed states. Detailed setup information of these examples are in Appendix D.

### 5.1.5 MARKOVIAN JUMP SYSTEMS

In addition to the systems described above, we introduce two Markovian jump systems (MJS), named MJS1 and MJS2, which contain both discrete switchings (or jumps) and continuous dynamics (Shi &

Li, 2015), as test beds for the proposed and baseline methods. The objective is to force the full state to zero. Both MJSs contain unstable subsystems and the dynamics change abruptly and randomly according to the switching signal, and thus are difficult to tackle for the model-free algorithms. Moreover, MJS2 contains an unstable subsystem that is not controllable, which makes it even harder to stabilize. More details on the examples could be found in Appendix E.

## 5.2 PERFORMANCE

We parameterize the policy and Lyapunov critic using deep neural networks. For each example, the hyperparameters including time horizon $N$ and DNN architectures selected to construct Lyapunov functions and DNN training parameters can be found in Appendix J. For both algorithms, the hyperparameters are tuned to reach their best performance. In each task, both LAC and SAC are trained for 10 times with random initialization, average total cost and its variance during training are demonstrated in Figure 1.

In the first three examples (see Figure 1(a)-(c)), SAC and LAC perform comparably in terms of the total cost at convergence and speed of convergence, while SPPO could converge in Cartpole and FetcheReach. In the Repressilator and MJS examples (see Figure 1(d,e,f)), SAC is not always able to find a policy that is capable of completing control objective, resulting in the bad average performance. On the contrary, LAC performs stably regardless of the random initialization.

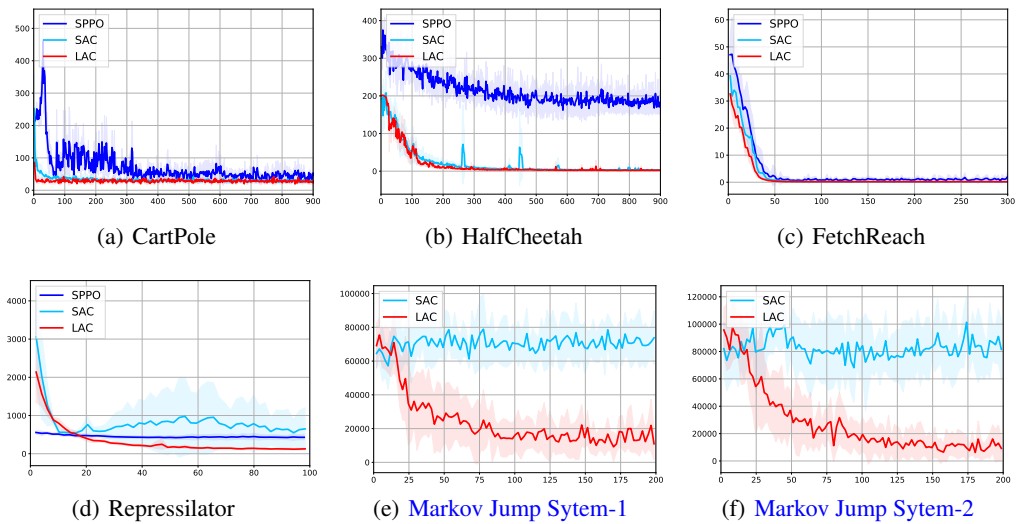

(a) CartPole        (b) HalfCheetah        (c) FetchReach

(d) Repressilator        (e) Markov Jump Sytem-1        (f) Markov Jump Sytem-2

Figure 1: Cumulative control performance comparison. The Y-axis indicates the total cost during one episode and the X-axis indicates the total time steps in thousand. The shadowed region shows the 1-SD confidence interval over 10 random seeds. Across all trials of training, LAC converges to stabilizing solution with comparable or superior performance compared with SAC and SPPO. The experiment on Complicated-Repressilator is deferred to Appendix F.

A distinguishing feature of stability assured policy is that it can force and sustain the state or tracking error to zero. This could be intuitively demonstrated by the state trajectories of closed-loop system. We evaluated this property of trained policies in the Repressilator, Complicated-Repressilator and two MJS examples. In our experiments, we found that the LAC agents stabilize the systems well in all tasks. All the state trajectories converge to the reference signal or equilibrium eventually (see Figure 11 (a,c) and Figure 12 (a,c)). On the contrary, without stability guarantee, the state trajectories either diverge (see Figure 11 b and Figure 12 d), or continuously oscillate around the reference trajectory or equilibrium (see Figure 11 d and Figure 12 b). Empirical results are deferred to Appendix F due to space limit.

## 5.3 CONVERGENCE

As shown in Figure 1, LAC converges stably in all experiments. Moreover, the convergence and validation of stability guarantee could also be checked by observing the value of Lagrange multipliers. When (2) is satisfied, $\lambda$ will continuously decrease until it becomes zero. Thus by checking the value

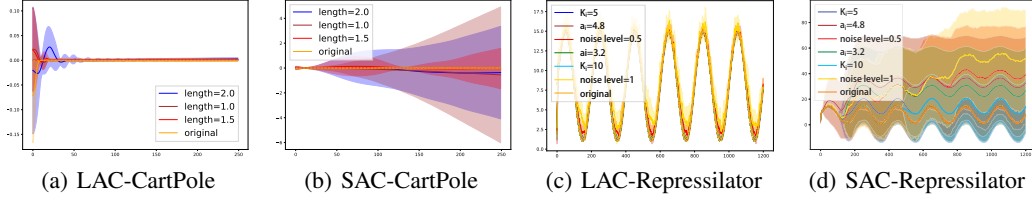

(a) Cartpole   (b) HalfCheetah   (c) FetchReach   (d) Repressilator   (e) MJS1   (f) MJS2

Figure 2: Value of Lagrange multiplier $\lambda$ during the training of LAC policies. The Y-axis indicates the value of $\lambda$ and the X-axis indicates the total time steps in thousand. The shadowed region shows the 1-SD confidence interval over 10 random seeds. The value of $\lambda$ gradually drops and becomes zero at convergence, which implies the satisfaction of stability condition.

and variation of $\lambda$, the satisfaction of stability condition during training and at convergence could be validated. In Figure 2, the value of $\lambda$ during training is demonstrated. Across all training trials in the experiments, $\lambda$ converges to zero eventually, which implies that the stability guarantee is valid. A detailed discussion on this is referred to Appendix G.

## 5.4 EVALUATION ON ROBUSTNESS AND GENERALIZATION

It is well-known that over-parameterized policies are prone to become overfitted to a specific training environment. The ability of generalization is the key to the successful implementation of the algorithm in an uncertain real-world environment. In this part, we first evaluate the robustness of policies in the presence of system parametric uncertainties and process noise. Then, we test the robustness of controllers against external disturbances. Finally, we evaluate whether the policy is generalizable by setting different reference signals. To make a fair comparison, we removed the policies that did not converge in SAC and only evaluate the ones that perform well during training. During testing, we found that SPPO appears to be prone to variation in the environment, thus the evaluation results are referred to Appendix H.

### 5.4.1 ROBUSTNESS TO DYNAMIC UNCERTAINTY

In this part, during the inference, we vary the system parameters in the model/simulator to evaluate the algorithm's robustness against dynamic uncertainty. In the example of CartPole, we vary the length of pole $l$. In the example of repressilator, we vary the promoter strength $a_i$ and dissociation rate $K_i$. Due to stochastic nature in gene expression, we also introduce uniformly distributed noise ranging from $[-\delta, \delta]$ (we indicate the *noise level* by $\delta$) to the dynamic of repressilator. The stabilization performance of CartPole and tracking performance of Repressilator by LAC and SAC in the varied environment is demonstrated in Figure 3.

(a) LAC-CartPole   (b) SAC-CartPole   (c) LAC-Repressilator   (d) SAC-Repressilator

Figure 3: State trajectories over time under policies trained by LAC and SAC and tested in the presence of parametric uncertainties and process noise, for CartPole and Repressilator. Solid line indicates the average trajectory and shadowed region for the 1-SD confidence interval. In (a) and (b), the pole length is varied during the inference. In (c) and (d), three parameters are selected to reflect the uncertainties in gene expression. The X-axis indicates the time and Y-axis shows the angle of pole in (a,b) and concentration of protein to be controlled in (c,d), respectively. Dashed line indicates the reference signal. The line in orange indicates the dynamic in original environment. For each curve, only the noted parameter is different with the original setting. We also show the curves in separate zoom-in view in Appendix I.1.

As shown in Figure 3(a) and (c), the policies trained by LAC are very robust to parametric uncertainties of different values and achieve high tracking precision in each case. On the other hand, though SAC performs well in the original environment (Figure 3(b) and (d)), it fails to track the reference signal in all of the varied environment.

### 5.4.2 ROBUSTNESS TO DISTURBANCES

An inherent property of a controller for stabilization is to enable the system to recover to the normal status from perturbations such as external forces and wind. To show this, we introduce persistent external disturbances with different magnitudes in each environment and observe the performance difference between policies trained by LAC and SAC. We also include LQR as the model-based baseline. In CartPole, the agent may fall over when interfered by an external force, ending the episode in advance. Thus in this task, we measure the robustness of controller through the death-rate, i.e., the probability of falling over after being disturbed. For other tasks where the episodes are always of the same length, we measure the robustness of controller by the variation in total cost. Under each disturbance magnitude, the policies are tested for 100 trials and the performance are shown in Figure 4.

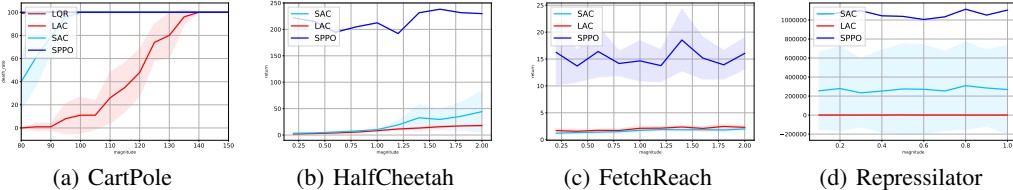

|        (a) CartPole        |        (b) HalfCheetah        |        (c) FetchReach        |        (d) Repressilator        |

Figure 4: Performance of policies trained by LAC, SAC and SPPO, along with controllers designed by LQR in the presence of persistent disturbances with different magnitudes. X-axis indicates the magnitude of the applied disturbance. For CartPole (a) the Y-axis indicates the probability of falling over and in other three examples (b)-(d) it indicates the total cost. Both policies are evaluated for 100 trials in each setting.

As shown in the Figure 4, the controller trained by LAC outperforms SAC and LQR by great extent when faced with external disturbances in CartPole and repressilator (lower death rate and total cost). In the repressilator example, the policies trained by SAC are extremely vulnerable to disturbances, this is potentially due to the existence of an unstable limit cycle in the uncontrolled dynamic (Strelkowa & Barahona, 2010). In HalfCheetah, SAC and LAC are both robust to small external disturbances while LAC is more reliable to larger ones. In FetchReach, SAC and LAC are comparable with maintaining a low cost in a great range of external disturbances. This is perhaps due to the manipulator's inherent robust mechanical design.

### 5.4.3 GENERALIZATION OVER DIFFERENT TRACKING REFERENCES

In this part, we introduce four different reference signals that are unseen during training in the repressilator example: sinusoids with periods of 150 (brown) and 400 (blue), and the constant reference of 8 (red) and 16 (green). We also show the original reference signal used for training (skyblue) as a benchmark. Reference signals are indicated in Figure 5 by the dashed line in respective colors. Both trained policies are evaluated to track each reference signal for 10 times, and the average dynamics of the target protein concentration are shown in Figure 5 with the solid line, while the variance of dynamic is indicated by the shadowed area.

As shown in Figure 5, the policies trained by LAC could generalize well to follow previously unseen reference signals with low deviation (dynamics are very close to the dashed lines), regardless of whether they are in the same mathematical form with the one used for training or not. On the other hand, though SAC tracks the original reference signal well after the unconverged training trials being removed (see the skyblue lines), it is still unable to follow some of the reference signals (see the brown line) and possesses larger variance than LAC when following others.

### 5.5 INFLUENCE OF DIFFERENT LYAPUNOV FUNCTION CANDIDATES AND STRUCTURES

In this part, we evaluate the influence of choosing different Lyapunov function candidates and network structures. First, we adopt candidates of different time horizon $N \in \{5, 10, 15, 20, \infty\}$ to train policies in the CartPole example, and compare their performance in terms of total cost and robustness. Both of the Lyapunov critics are parameterized by $L(s) = \phi(s)^T \phi(s)$ where $\phi(s)$ is a neural network with $m$ dimensional output. Here, $N = \infty$ implies using value function as Lyapunov candidate. For evaluation of robustness, we apply an impulsive force $F$ at $100_{th}$ instant and observe the death-rate of trained policies. The results are demonstrated in Figure 6 (a,b). Then we fix the

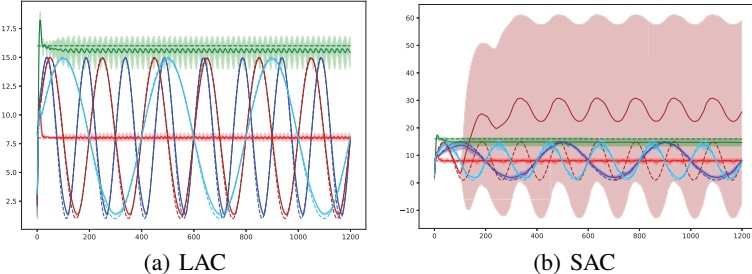

(a) LAC      (b) SAC

Figure 5: State trajectories under policies trained by LAC and SAC when tracking different reference signals. Solid line indicates the average trajectory and shadowed region for the 1-SD confidence interval. The X-axis indicates the time and Y-axis shows the concentration of protein to be controlled. Dashed lines in different colors are the different reference signals: sinusoid with period of 150 (brown); sinusoid with period of 200 (skyblue);sinusoid with period of 400 (blue); constant reference of 8 (red); constant reference of 16 (green). We also show the curves in separate zoom-in view in Appendix I.2 .

horizon of candidates to be $N = 5$ but vary the structures of Lyapunov critic, and compare their performance using the same metric as described above. More specifically, these different structures are: $L(s) = \sum_{j=1}^{m} \phi_j^4(s)$ (LAC-biquad); $L(s) = \sum_{j=1}^{m} |\phi(s)|$ (LAC-abs); $L(s) = \phi(s)^T \phi(s)$(LAC-quad).

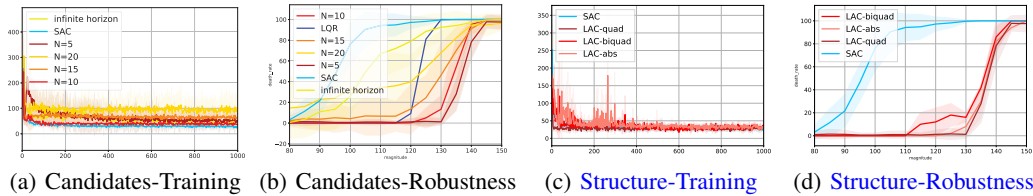

(a) Candidates-Training (b) Candidates-Robustness (c) Structure-Training (d) Structure-Robustness

Figure 6: Influence of different Lyapunov function candidates and network structures. In (a) and (c), the Y-axis indicates total cost of policies during training by LAC with Lyapunov function candidates of different length of horizon $N$ and structures, and the X-axis indicates the total time steps in thousand. (b) and (d) shows the death-rate of policies in the presence of instant impulsive force $F$ ranging from 80 to 150 Newton.

As shown in Figure 6, in the CartPole environment, both choices of Lyapunov candidates converge fast and achieve comparable total cost at convergence. However, in terms of robustness, the different choices of $N$ play an important role. As observed in Figure 6 (b), the robustness of controller decreases as the time horizon $N$ increases. On the other hand, LAC with different structures converge well and possesses similar robustness to impulsive forces. This further proves that our framework allows for a general class of Lyapunov functions, as long as the function is semi-positive definite. Besides, it is interesting to observe that LQR is more robust than SAC when faced with instant impulsive disturbance.

## 6 CONCLUSIONS

In this paper, we proposed a model-free approach for analyzing the stability of discrete-time nonlinear stochastic systems modeled by Markov decision process, by employing the Lyapunov function from control theory. Based on the theoretical result, a practical algorithm for designing stability assured controllers for the stabilization and tracking problems. We evaluated the proposed method in various examples and show that our method achieves not only comparable or superior performance compared with the state-of-the-art RL algorithm but also outperforms impressively in terms of robustness to uncertainties and disturbances.

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

# Appendix

## A PROOF OF THEOREM 1

The existence of sampling distribution $\mu_\pi(s)$ is guaranteed by the existence of $q_\pi(s)$ (Assumption 1). Since the sequence $\{P(s|\rho,\pi,t), t \in \mathbb{Z}_+\}$ converges to $q_\pi(s)$ as $t$ approaches $\infty$, then by the Abelian theorem, the sequence $\{\frac{1}{N}\sum_{t=0}^N P(s|\rho,\pi,t), N \in \mathbb{Z}_+\}$ also converges and $\mu_\pi(s) = q_\pi(s)$. Combined with the form of $\mu_\pi$, Eq.(2) infers that

$$\int_{\mathcal{S}} \lim_{N\to\infty} \frac{1}{N}\sum_{t=0}^N P(s|\rho,\pi,t)(\mathbb{E}_{P_\pi(s'|s)}L(s') - L(s))\mathrm{d}s \le -\alpha_3 \mathbb{E}_{s\sim q_\pi}c_\pi(s) \tag{A.1}$$

First, on the left hand-side, according to Eq.(1), $L(s) \le \alpha_2 c_\pi(s)$ for all $s \in \mathcal{S}$ and consider that $P(s|\rho,\pi,t) \le 1$,
$$P(s|\rho,\pi,t)L(s) \le \alpha_2 c_\pi(s), \forall s \in \mathcal{S}, \forall t \in \mathbb{Z}_+$$

On the other hand, the sequence $\{\frac{1}{N}\sum_{t=0}^N P(s|\rho,\pi,t)L(s), N \in \mathbb{Z}_+\}$ converges pointwise to the function $q_\pi(s)L(s)$. According to the Lebesgue's Dominated convergence theorem(Royden, 1968), if a sequence $f_n(s)$ converges pointwise to a function $f$ and is dominated by some integrable function $g$ in the sense that,
$$|f_n(s)| \le g(s), \forall s \in \mathcal{S}, \forall n$$
Then
$$\lim_{n\to\infty} \int_{\mathcal{S}} f_n(s)\mathrm{d}s = \int_{\mathcal{S}} \lim_{n\to\infty} f_n(s)\mathrm{d}s$$
Thus the left hand side of Eq.(A.1)

$$\int_{\mathcal{S}} \lim_{N\to\infty} \frac{1}{N}\sum_{t=0}^N P(s|\rho,\pi,t)(\int_{\mathcal{S}} P_\pi(s'|s)L(s')\mathrm{d}s' - L(s))\mathrm{d}s$$
$$= \lim_{N\to\infty} \frac{1}{N}(\sum_{t=1}^{N+1} \mathbb{E}_{P(s|\rho,\pi,t)}L(s) - \sum_{t=0}^N \mathbb{E}_{P(s|\rho,\pi,t)}L(s))$$
$$= \lim_{N\to\infty} \frac{1}{N}\left(\mathbb{E}_{P(s|\rho,\pi,N+1)}L(s) - \mathbb{E}_{\rho(s)}L(s)\right)$$

Thus taking the relations above into consideration, Eq.(A.1) infers

$$\lim_{N\to\infty} \frac{1}{N}\left(\mathbb{E}_{P(s|\rho,\pi,N+1)}L(s) - \mathbb{E}_{\rho(s)}L(s)\right) \le -\alpha_3 \lim_{t\to\infty} \mathbb{E}_{P(s|\rho,\pi,t)}c_\pi(s) \tag{A.2}$$

Since $\mathbb{E}_{\rho(s)}L(s)$ is a finite value and $L$ is semi-positive definite, it follows that

$$\lim_{t\to\infty} \mathbb{E}_{P(s|\rho,\pi,t)}c_\pi(s) \le \lim_{N\to\infty} \frac{1}{N}(\frac{1}{\alpha_3}\mathbb{E}_{\rho(s)}L(s)) = 0 \tag{A.3}$$

Suppose that there exists a state $s_0 \in \{s_0|c_\pi(s_0) \le b\}$ such $\lim_{t\to\infty} \mathbb{E}_{P(s|s_0,\pi,t)}c_\pi(s) = c, c > 0$ or $\lim_{t\to\infty} \mathbb{E}_{P(s|s_0,\pi,t)}c_\pi(s) = \infty$. Consider that $\rho(s_0) > 0$ for all starting states in $\{s_0|c_\pi(s_0) \le b\}$ (Assumption 2), then $\lim_{t\to\infty} \mathbb{E}_{s_t\sim P(\cdot|\pi,\rho)}c_\pi(s_t) > 0$, which is contradictory with Eq.(A.3). Thus $\forall s_0 \in \{s_0|c_\pi(s_0) \le b\}$, $\lim_{t\to\infty} \mathbb{E}_{P(s|s_0,\pi,t)}c_\pi(s) = 0$. Thus the system is stable in mean cost by Definition 1.

## B    PSEUDO CODE OF ALGORITHM

---

**Algorithm 1** Lyapunov-based Actor-Critic (LAC)

---

Initialize replay buffer $\mathcal{D}$ and Lagrange multiplier $\lambda$, $\beta$;

Randomly initialize Lyapunov critic network $L_c(s, a)$, actor $\pi(a|s)$ with parameters $\phi_{L_c}$, $\theta$;

Initialize the parameters of target network with $\overline{\theta} \leftarrow \theta$;

**for** each iteration **do**

    Sample $s_0$ according to $\rho$;

    **for** each time step **do**

        Sample $a_t$ from $\pi(s)$ and step forward;

        Observe $s_{t+1}$, $c_t$ and store $(s_t, a_t, c_t, s_{t+1})$ in $\mathcal{D}$;

    **end for**

    **for** each update step **do**

        Sample minibatches of transitions from $\mathcal{D}$ and update $L_c$, $\pi$, Lagrange multipliers with gradients;

        Update the target networks:

$$\overline{\theta} \leftarrow \tau\theta + (1 - \tau)\overline{\theta}$$

    **end for**

**end for**

---

## C FURTHER EXPERIMENT SETUP

We setup the experiment using OpenAi Gym (Brockman et al., 2016). A snapshot of environments can be found in Figure 7.

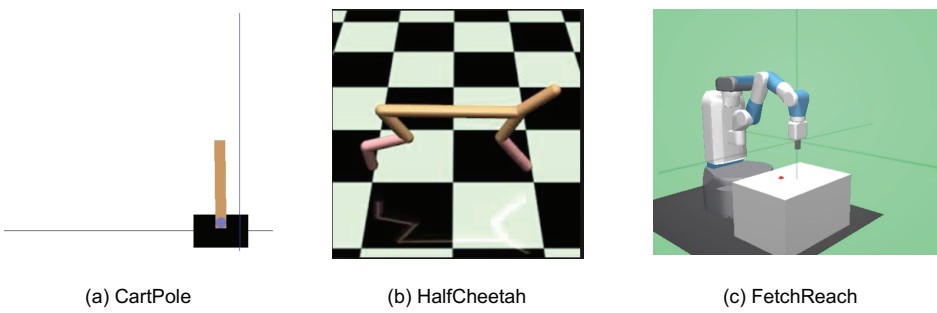

(a) CartPole        (b) HalfCheetah        (c) FetchReach

Figure 7: Snapshot of environments using OpenAI Gym.

### C.1 CARTPOLE

In this experiment, the controller is to sustain the pole vertically at a target position $x = 0$. This is modified version of CartPole in Brockman et al. (2016) with continuous action space. The action is the horizontal force applied on the cart ($a \in [-20, 20]$). $x_{\text{threshold}}$ and $\theta_{\text{threshold}}$ represents the maximum of position and angle, respectively, $x_{\text{threshold}} = 10$ and $\theta_{\text{threshold}} = 20°$. The controller dies if $|x| > x_{\text{threshold}}$ or $|\theta| > \theta_{\text{threshold}}$ and the episodes end in advance. Cost function $r = (\frac{x}{x_{\text{threshold}}})^2 + 20 * (\frac{\theta}{\theta_{\text{threshold}}})^2$. The episodes are of length 250. For robustness evaluation in Section 5.4, we apply an impulsive disturbance force $F$ on the cart every 20 seconds, of which the magnitude ranges from 80 to 150 and the direction is opposite to the direction of control input. In Section 5.5, the impulsive disturbance has the same magnitude range and direction with that in Section 5.4, but only applied once at instant $t = 100$.

### C.2 HALFCHEETAH

HalfCheetah is a modified version of that in Gym's robotics environment (Brockman et al., 2016). The task is to control a HalfCheetah (a 2-legged simulated robot) to run at the speed of $1 \ m/s$. The reward is $r = (v - 1)^2$ where $v$ is the forward speed of the HalfCheetah. The control input is the torque applied on each joint, ranging from -1 to 1. The episodes are of length 200.

For robustness evaluation in Section 5.4, we apply an impulsive disturbance torque on each joint every 20 seconds, of which the magnitude ranges from 0.2 to 2.0 and the direction is opposite to the direction of control input.

### C.3 FETCHREACH-V1

We modify the FetchReach in Gym's robotics environment (Brockman et al., 2016) to a cost version, where the controller is expected to control manipulator's end effector to reach a random goal position. The cost is designed as $c = d$, where $d$ is the distance between goal and end-effector. The control input is the torque applied on each joint, ranging from -1 to 1. The episodes are of length 200.

For robustness evaluation in Section 5.4, we apply an impulsive disturbance torque on each joint every 20 seconds, of which the magnitude ranges from 0.2 to 2.0 and the direction is opposite to the direction of control input.

# D    SYNTHETIC BIOLOGY GENE REGULATORY NETWORKS

Since this system considered here is in nano-scale whose physical property is different from the ones considered in Section C and the system exhibit interesting oscillatory behavior, we illustrate this example separately in this section.

## D.1    MATHEMATICAL MODEL OF REPRESSILATOR

In this example, we consider a classical dynamical system in systems/synthetic biology, the repressilator, which we use to illustrate the reference tracking problem at hand. The repressilator is a synthetic three-gene regulatory network where the dynamics of mRNAs and proteins follow an oscillatory behavior (Elowitz & Leibler, 2000). A discrete-time mathematical description of the repressilator, which includes both transcription and translation dynamics, is given by the following set of discrete-time equations:

$$
\begin{aligned}
x_1(t+1) &= x_1(t) + dt \cdot \left[ -\gamma_1 x_1(t) + \frac{a_1}{K_1 + x_6^2(t)} + u_1(t) \right] + \xi_1(t), \\
x_2(t+1) &= x_2(t) + dt \cdot \left[ -\gamma_2 x_2(t) + \frac{a_2}{K_2 + x_4^2(t)} + u_2(t) \right] + \xi_2(t), \\
x_3(t+1) &= x_3(t) + dt \cdot \left[ -\gamma_3 x_3(t) + \frac{a_3}{K_3 + x_5^2(t)} + u_3(t) \right] + \xi_3(t), \\
x_4(t+1) &= x_4(t) + dt \cdot \left[ -c_1 x_4(t) + \beta_1 x_1(t) \right] + \xi_4(t), \\
x_5(t+1) &= x_5(t) + dt \cdot \left[ -c_2 x_5(k) + \beta_2 x_2(t) \right] + \xi_5(t), \\
x_6(t+1) &= x_6(t) + dt \cdot \left[ -c_3 x_6(t) + \beta_3 x_3(t) \right] + \xi_6(t).
\end{aligned}
\tag{D.1}
$$

Here, $x_1, x_2, x_3$ (resp. $x_4, x_5, x_6$) denote the concentrations of the mRNA transcripts (resp. proteins) of genes 1, 2, and 3, respectively. $\xi_i, \forall i$ are i.i.d. uniform noise ranging from $[-\delta, \delta]$, i.e., $\xi_i \sim \mathcal{U}(-\delta, \delta)$. During training, $\delta = 0$ and for evaluation $\delta$ is set to 0.5 and 1 respectively in Section 5.4. $a_1, a_2, a_3$ denote the maximum promoter strength for their corresponding gene, $\gamma_1, \gamma_2, \gamma_3$ denote the mRNA degradation rates, $c_1, c_2, c_3$ denote the protein degradation rates, $\beta_1, \beta_2, \beta_3$ denote the protein production rates, and $K_1, K_2, K_3$ are the dissociation constants. The set of equations in Eq.(D.1) corresponds to a topology where gene 1 is repressed by gene 2, gene 2 is repressed by gene 3, and gene 3 is repressed by gene 1. $dt$ is the discretization time step.

In practice, only the protein concentrations are observed and given as readouts,for instance via fluorescent markers (e.g., green fluorescent protein, GFP or red fluorescent protein, mCherry). The control scheme $u_i$ will be implemented by light control signals which can induce the expression of genes through the activation of their photo-sensitive promoters. To simplify the system dynamics and as it is usually done for the repressilator model (Elowitz & Leibler, 2000), we consider the corresponding parameters of the mRNA and protein dynamics for different genes to be equal. More background on mathematical modeling and control of synthetic biology gene regulatory networks can be referred to Strelkowa & Barahona (2010); Sootla et al. (2013). In this example, the parameters are as follows:

$$
\forall i: \ K_i = 1, a_i = 1.6, \gamma_i = 0.16, \beta_i = 0.16, c_i = 0.06, dt = 1
$$

In Fig8, a single snapshot of the state temporal evolution without $\xi$ is depicted. We uniformly initialized between 0 to 5, i.e., $x_i(0) \sim \mathcal{U}(0, 5)$, which is the range we train the policy in Section 5, persistent oscillatory behavior are also exhibiting similar to the snapshot in Fig 8.

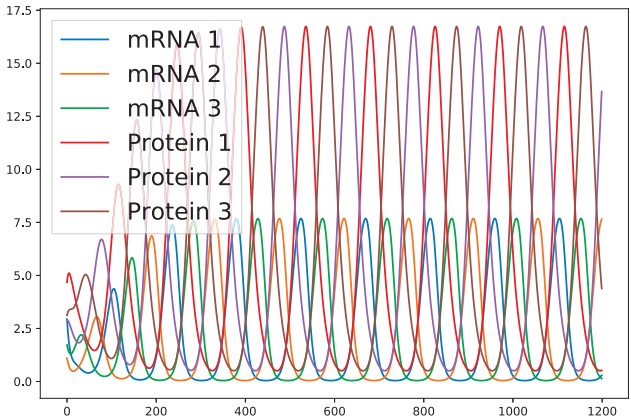

Figure 8: A snapshot of natural oscillatory behaviour of a repressilator system consisting of 3 genes. The oscillations have a period of approximately 150 arbitrary time units. The X-axis denotes time and Y-axis denotes value/concentration of each state.

### D.2 COMPLICATED REPRESSILATOR

To further evaluate the performance of different algorithms, we additionally include a more complicated gene regulatory network, which is composed of 4 genes instead of 3. Such a network with 4 genes would posses an unstable oscillatory behaviour, as shown in Figure 9, making it even harder to stabilize. The discrete-time mathematical description of the complicated repressilator is given by the following set of discrete-time equations:

$$
\begin{aligned}
x_1(t+1) &= x_1(t) + dt \cdot \left[ -\gamma_1 x_1(t) + \frac{a_1}{K_1 + x_8^2(t)} + u_1(t) \right] + \xi_1(t), \\
x_2(t+1) &= x_2(t) + dt \cdot \left[ -\gamma_2 x_2(t) + \frac{a_2}{K_2 + x_5^2(t)} + u_2(t) \right] + \xi_2(t), \\
x_3(t+1) &= x_3(t) + dt \cdot \left[ -\gamma_3 x_3(t) + \frac{a_3}{K_3 + x_6^2(t)} + u_3(t) \right] + \xi_3(t), \\
x_4(t+1) &= x_4(t) + dt \cdot \left[ -\gamma_4 x_4(t) + \frac{a_4}{K_4 + x_7^2(t)} + u_4(t) \right] + \xi_4(t), \\
x_5(t+1) &= x_5(t) + dt \cdot \left[ -c_1 x_5(t) + \beta_1 x_1(t) \right] + \xi_5(t), \\
x_6(t+1) &= x_6(t) + dt \cdot \left[ -c_2 x_6(k) + \beta_2 x_2(t) \right] + \xi_6(t), \\
x_7(t+1) &= x_7(t) + dt \cdot \left[ -c_3 x_7(t) + \beta_3 x_3(t) \right] + \xi_7(t). \\
x_8(t+1) &= x_8(t) + dt \cdot \left[ -c_4 x_8(t) + \beta_4 x_4(t) \right] + \xi_8(t).
\end{aligned}
\tag{D.2}
$$

Here, $x_1, x_2, x_3, x_4$ (resp. $x_5, x_6, x_7, x_8$) denote the concentrations of the mRNA transcripts (resp. proteins) of genes 1, 2, 3 and 4, respectively.

The parameters are as follows,

$$
\forall i \in \{1, 2, 3, 4\} : \ K_i = 1, a_i = 1.6, \gamma_i = 0.16, \beta_i = 0.16, c_i = 0.06, dt = 1
$$

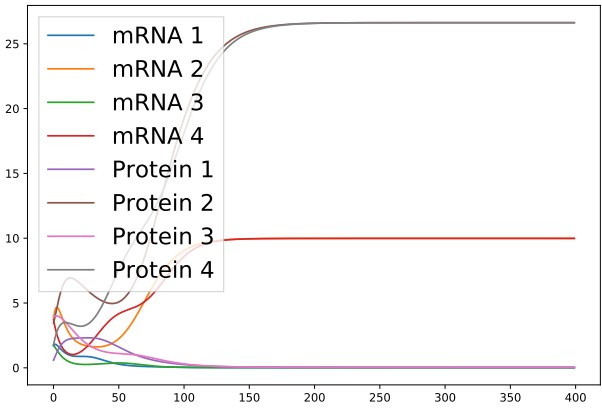

Figure 9: A snapshot of natural behaviour of a repressilator system consisting of 4 genes. The X-axis denotes time and Y-axis denotes value/concentration of each state.

# E MARKOVIAN JUMP SYSTEMS

In addition to the systems described in Section C and Section D, we introduce two Markovian jump systems (MJS), which contain both discrete switchings (or jumps) and continuous dynamics (Shi & Li, 2015), as test beds for the proposed and baseline methods. Specifically, we borrow two simple examples from the linear case of MJS, i.e. Markovian jump linear systems (MJLS). The dynamic of the MJLS could be described by the following state space model,

$$x_{k+1} = A_{\sigma_k} x_k + B_{\sigma_k} u_k \tag{E.1}$$

where $x_k$ and $u_k$ are the state and control inputs respectively; $A_{\sigma_k}$ and $B_{\sigma_k}$ are the parameter matrices. $\sigma_k$ is the switching signal governing the switching of subsystems, which takes value in a finite set $\{1, \ldots, N\}$ where $N$ is the number of subsystems. In MJS, the value of $\sigma$ is governed by a Markov process, $\sigma_{k+1} \sim P(\sigma_{k+1}|\sigma_k)$. The task of the agent is to stabilize the system and cost function is $c(x) = \|x\|_2$.

For the first MJS named MJS1, the parameter matrices are given as follow,

$$
\begin{aligned}
A_1 &= \rho \begin{bmatrix} -0.3672 & 0.7038 \\ -1.8462 & 2.0094 \end{bmatrix}, & B_1 &= \begin{bmatrix} -1 \\ 1 \end{bmatrix}, \\
A_2 &= \rho \begin{bmatrix} 0.3468 & 0.6324 \\ -0.7774 & 1.1872 \end{bmatrix}, & B_2 &= \begin{bmatrix} -1 \\ 1 \end{bmatrix}, \\
A_3 &= \rho \begin{bmatrix} -0.3468 & 0.6324 \\ -0.7774 & 1.1872 \end{bmatrix}, & B_3 &= \begin{bmatrix} 0 \\ 1 \end{bmatrix}
\end{aligned}
\tag{E.2}
$$

where $\rho = 1.3$ and the transition probability of switching signal at each instant is uniformly distributed across all three modes. Among the three subsystems, subsystem 1 and 2 are unstable without control input.

To further make the task more difficult, we include a second MJS system, MJS2, of which the parameter is given as

$$
\begin{aligned}
A_1 &= \rho \begin{bmatrix} -0.4227 & 0.7710 \\ -1.1600 & -0.6912 \end{bmatrix}, & B_1 &= \begin{bmatrix} 1 \\ 2 \end{bmatrix}, \\
A_2 &= \rho \begin{bmatrix} -0.5084 & 0.4536 \\ 1.0901 & -0.7266 \end{bmatrix}, & B_2 &= \begin{bmatrix} 0 \\ 0 \end{bmatrix}, \\
A_3 &= \rho \begin{bmatrix} -0.4772 & 0.7313 \\ 1.3938 & -0.7266 \end{bmatrix}, & B_3 &= \begin{bmatrix} 1 \\ 2 \end{bmatrix}
\end{aligned}
\tag{E.3}
$$

where $\rho = 0.859$ and the switching signal is also uniformly distributed. Note that the subsystem 2 and 3 are unstable without control input. Moreover, subsystem 2 in uncontrollable, since the control input cannot effect the system dynamic under this mode.

# F    FURTHER VALIDATION OF STABILITY GUARANTEE

In this part, a further comparison between the stability-assured method (LAC) and that without such guarantee (SAC) is made, by demonstrating the closed-loop system dynamic with the trained policies (in the two Repressilator and two MJS examples).

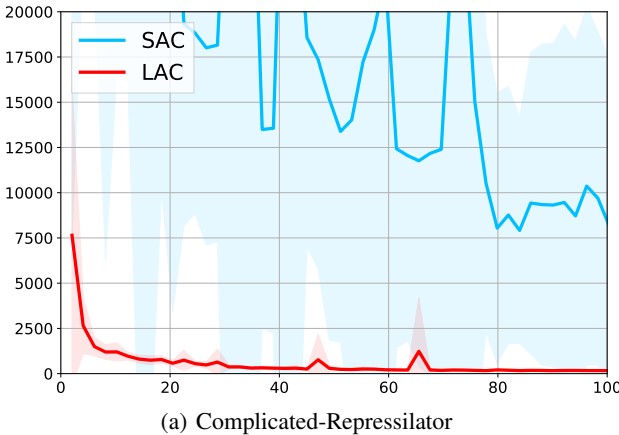

(a) Complicated-Repressilator

Figure 10: Cumulative control performance comparison. The Y-axis indicates the total cost during one episode and the X-axis indicates the total time steps in thousand. The shadowed region shows the 1-SD confidence interval over 10 random seeds. Across all trials of training, LAC converges to stable solution with comparable or superior performance compared with SAC.

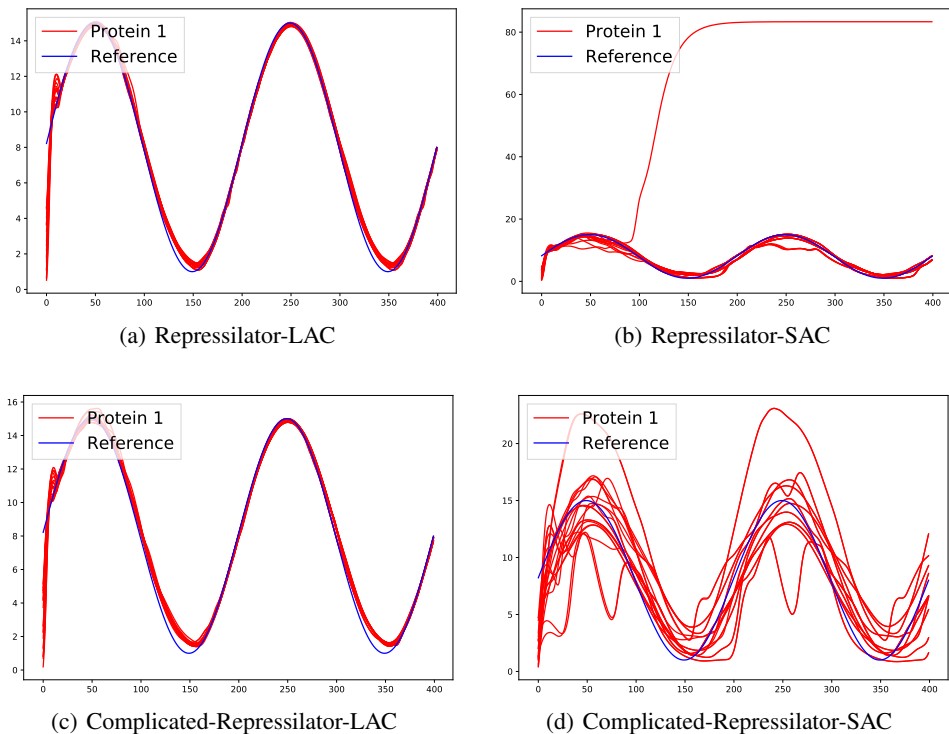

(a) Repressilator-LAC

(b) Repressilator-SAC

(c) Complicated-Repressilator-LAC

(d) Complicated-Repressilator-SAC

Figure 11: State trajectories over time under policies trained by LAC and SAC in the Repressilator and Complicated Repressilator. In each experiment, the policies are tested over 20 random initial states and all the resulting trajectories are displayed above. The X-axis indicates the time and Y-axis shows the concentration of Protein 1.

As shown in the figures, without stability guarantee, the state trajectories either diverge (see Figure 11 b and Figure 12 d), or continuously oscillate around the reference trajectory or equilibrium (see

Figure 11 d and Figure 12 b). In the MJS examples, the trajectories even diverge to or oscillate in unacceptable magnitude (1e7 and 1e10). Contrarily, the stability assured method stabilizes the system well in all tasks (i.e. the state trajectories converge to the reference signal or equilibrium). In the MJS examples, though temporal oscillation occurs in some of the trials due to the existence of unstable uncontrollable subsystems, eventually all of the trajectories are stabilized.

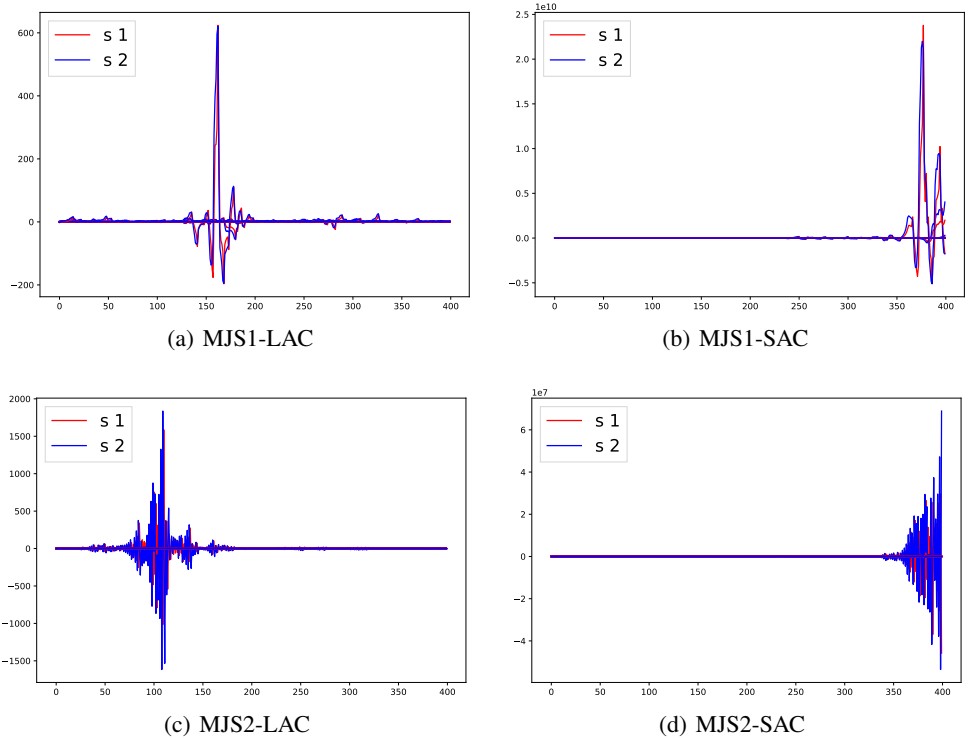

(a) MJS1-LAC          (b) MJS1-SAC

(c) MJS2-LAC          (d) MJS2-SAC

Figure 12: State trajectories over time under policies trained by LAC and SAC in the two Markovian jump systems. In each experiment, the policies are tested over 20 random initial states and all the resulting trajectories are displayed above. The X-axis indicates the time and Y-axis shows the value of states.

## G    FURTHER VALIDATION OF STABILITY GUARANTEE

In addition to the evaluation of stability in terms of system dynamic in previous sections, this part presents a more direct approach for validating the satisfaction of stability condition (2). The Algorithm 1 aims to solve the dual problem of the original policy optimization problem, i.e. solving the following min-max problem,

$$\max_{\lambda, \beta} \min_{\theta} J(\pi) \tag{G.1}$$

where $\lambda$ and $\beta$ are positive Lagrange multipliers and updated by gradient ascent. When (2) is satisfied, $\lambda$ will continuously decrease until it becomes zero. Thus by checking the value and variation of $\lambda$, the satisfaction of stability condition during training and at convergence could be validated. More specifically, the decline of $\lambda$ implies the satisfaction of stability condition at that update; $\lambda$ only converges to zero if the stability condition is assured.

Clipping the maximum value of $\lambda$ is necessary, in case that $\lambda$ grows too much due to the violation of stability condition during the early training stage, resulting in the inappropriate step length for the policy update. Clipping is a useful technique to prevent instability of optimization, especially in gradient-based methods, see Schulman et al. (2017); Bengio et al. (2013); Bello et al. (2017); Wang et al. (2015). Conversely, when the stability condition is satisfied, $\lambda$ quickly drops and helps convergence of the algorithm, which inherently prevents overfitting and enhances robustness and generalization.

# H    ROBUSTNESS AND GENERALIZATION EVALUATION OF SPPO

In this part, we evaluate the robustness and generalization ability of policies trained by SPPO in the same. First, the robustness of the policies is tested by perturbing the parameters and adding noise in the Cartpole and Repressilator environment, as described in Section 5.4.1. Generalization of the policies is evaluated by setting reference signals that are unseen during training. State trajectories of the above experiments are demonstrated in Figure 13 and Figure 14, respectively. As demonstrated in the figures, the SPPO policies could hardly deal with previously unseen uncertainty or reference signals, and failed in all of the Repressilator experiments.

The SPPO algorithm is originally developed for the control tasks with safety constraints, i.e. keeping the expectation of discounted cumulative safety cost below a certain threshold. Though Lyapunov method is exploited, the approach is not aimed at providing stability guarantee.

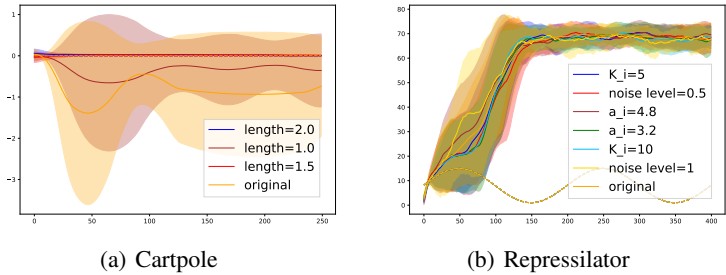

(a) Cartpole                    (b) Repressilator

Figure 13: State trajectories over time under policies trained by SPPO and tested in the presence of parametric uncertainties and process noise, for CartPole and Repressilator. The setting of the uncertainty is the same as in Section 5.4.1.

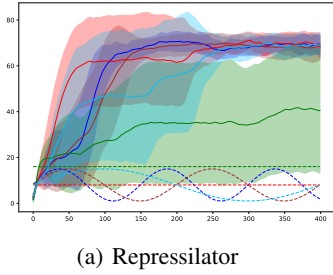

(a) Repressilator

Figure 14: State trajectories under policies trained by SPPO when tracking different reference signals. The setting of the uncertainty is the same as in Section 5.4.3.

# I  ZOOM-IN VIEWS

## I.1  ZOOM-IN VIEW OF FIGURE 3

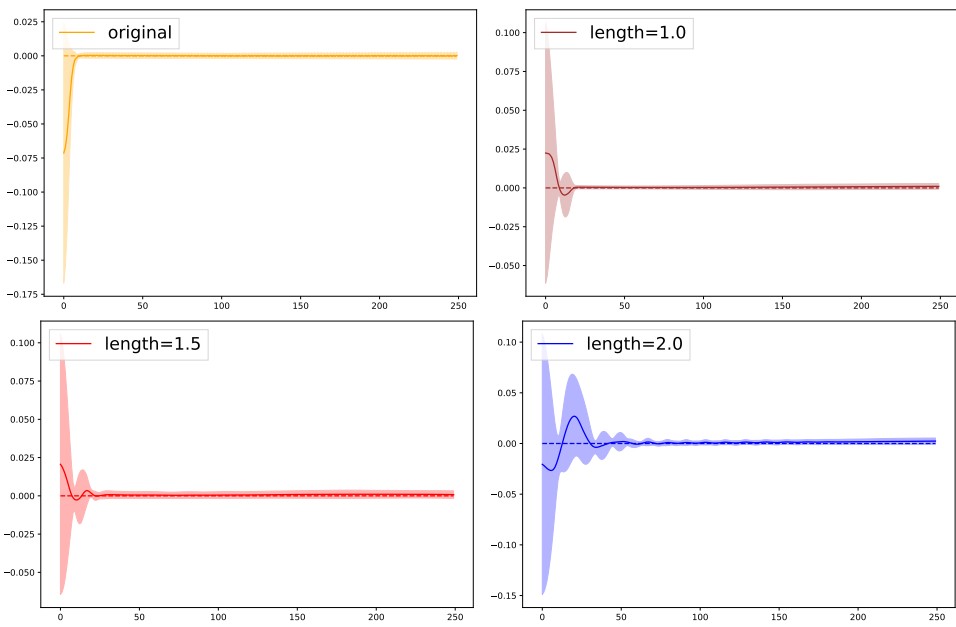

Figure 15: Zoom-in view of Figure 3 (a)

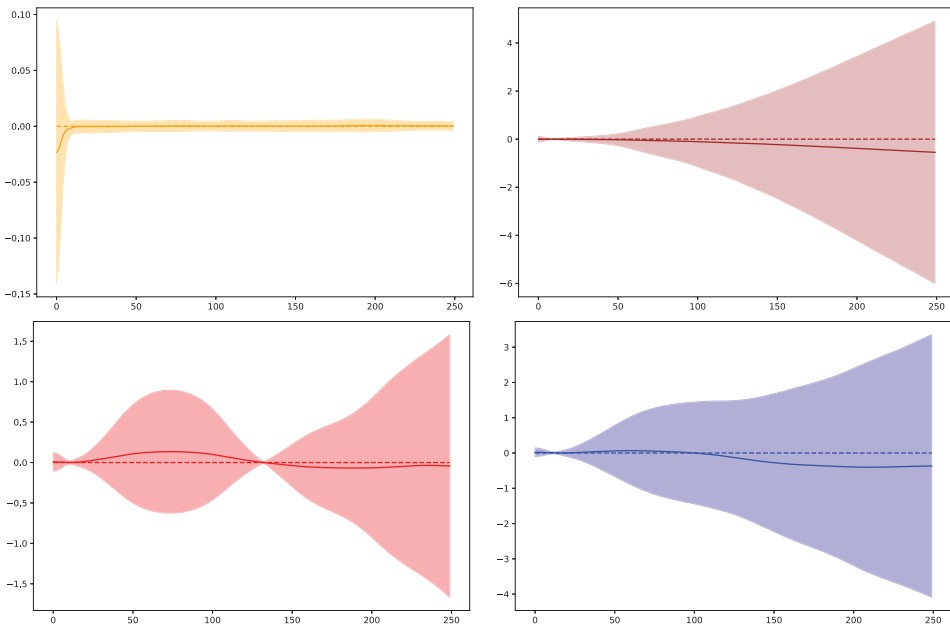

Figure 16: Zoom-in view of Figure 3 (b)

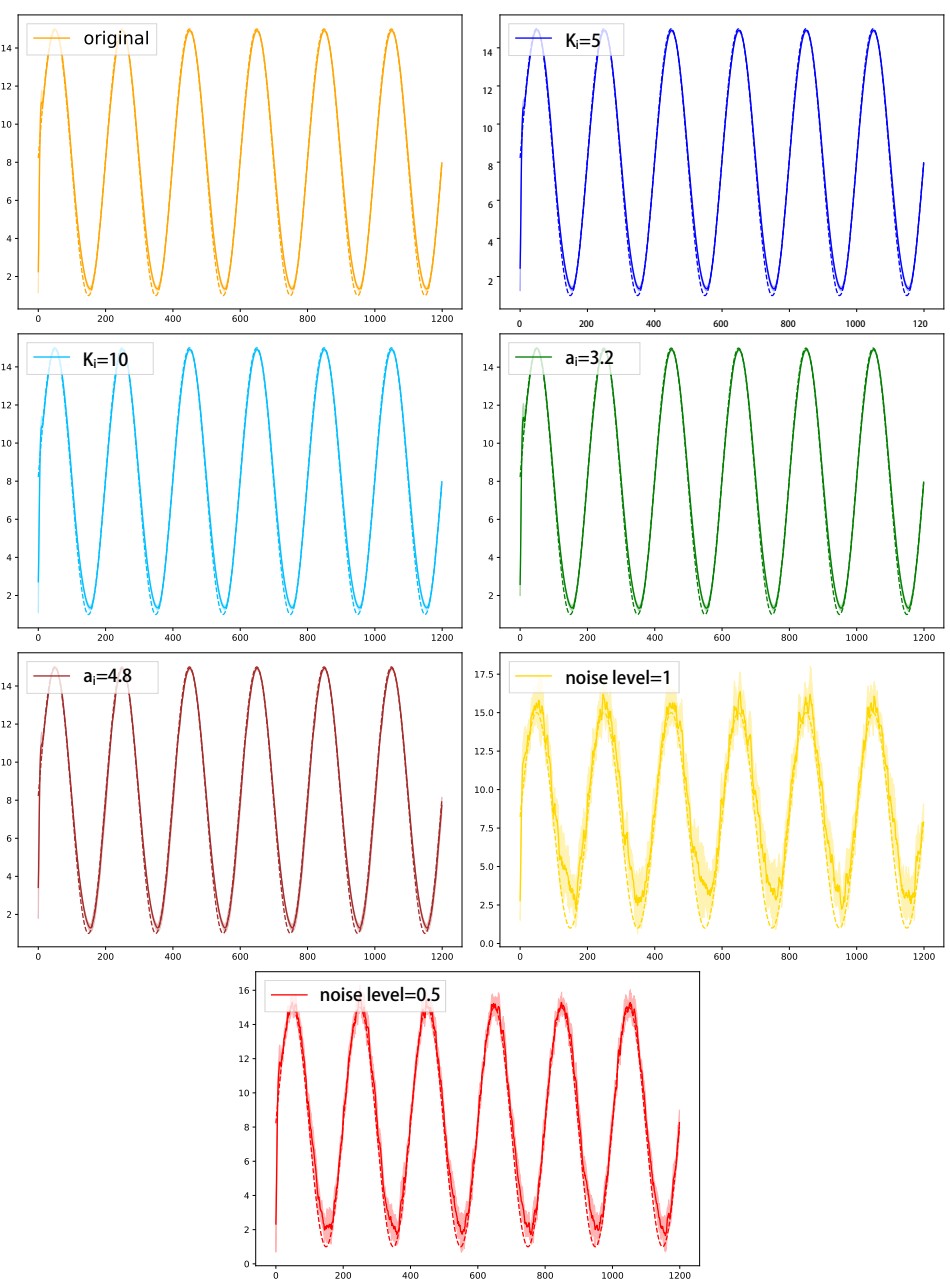

Figure 17: Zoom-in view of Figure 3 (c)

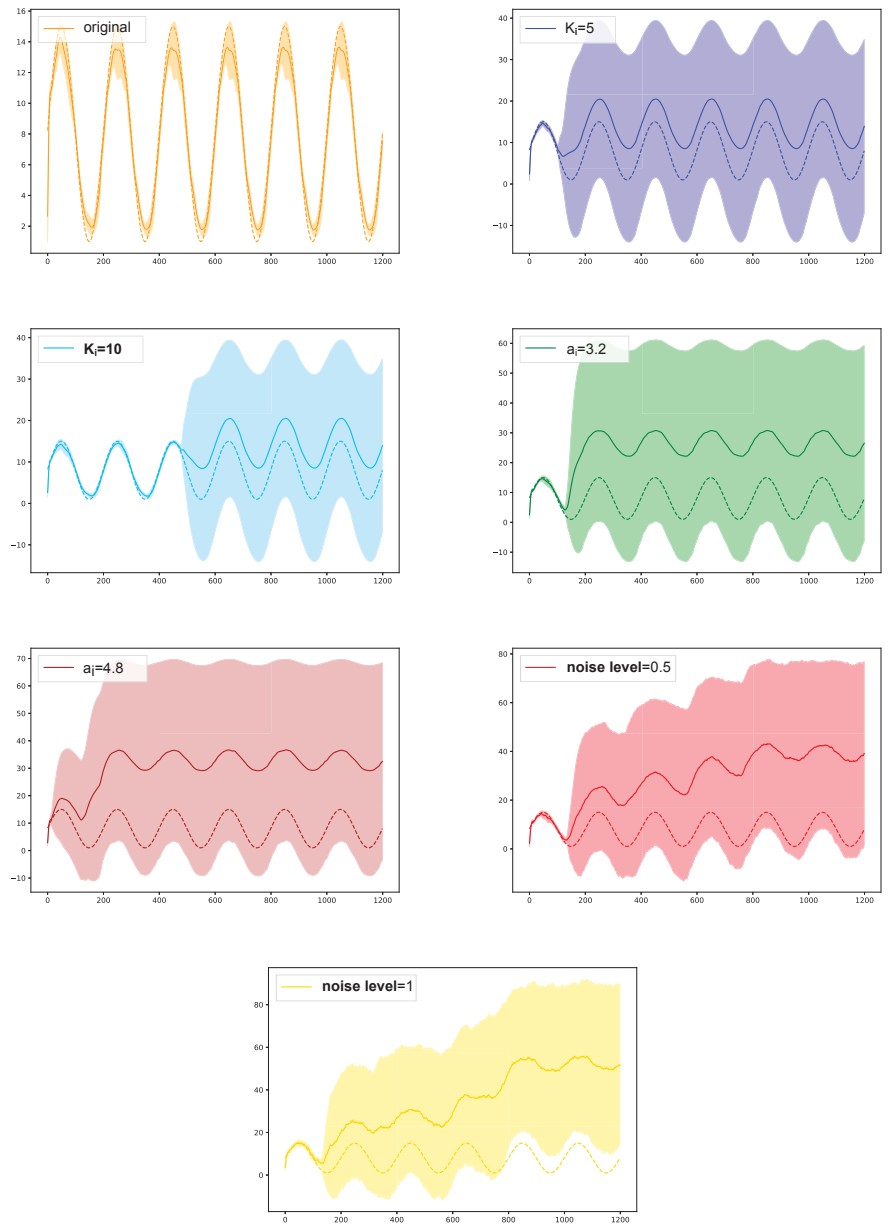

Figure 18: Zoom-in view of Figure 3 (d)

## I.2 ZOOM-IN VIEW OF FIGURE 5

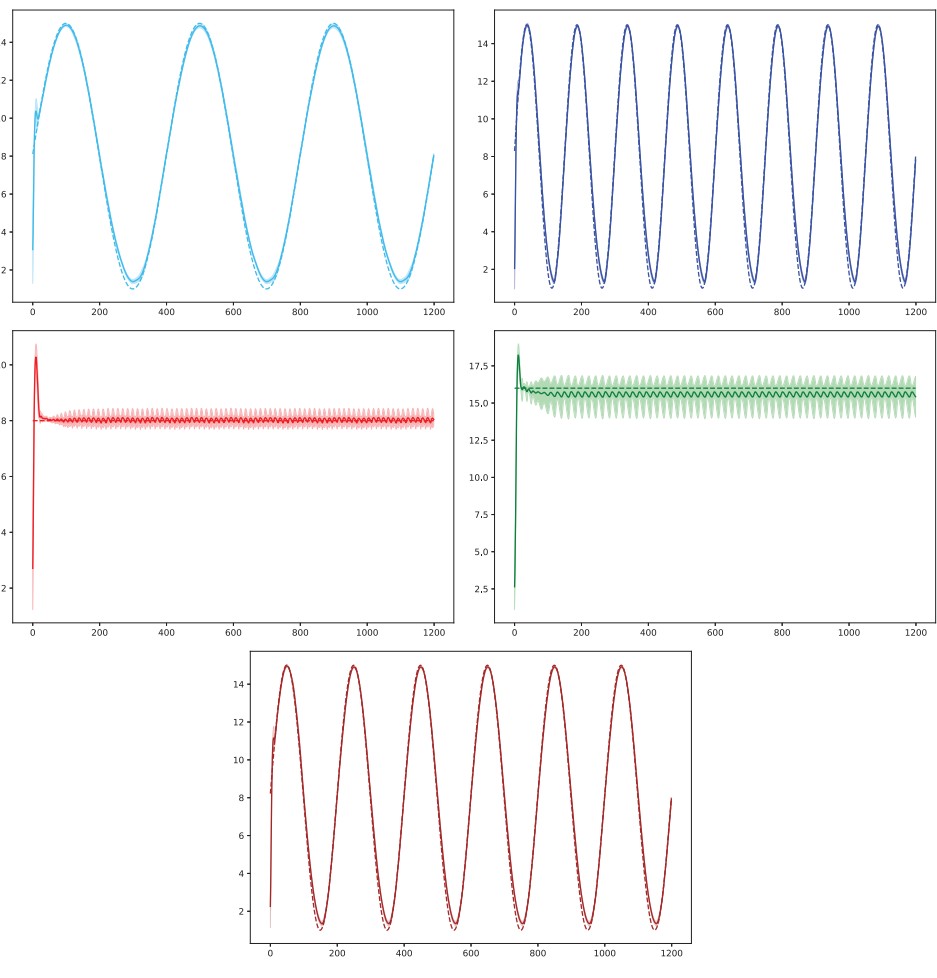

Figure 19: Zoom-in view of Figure 5 (a)

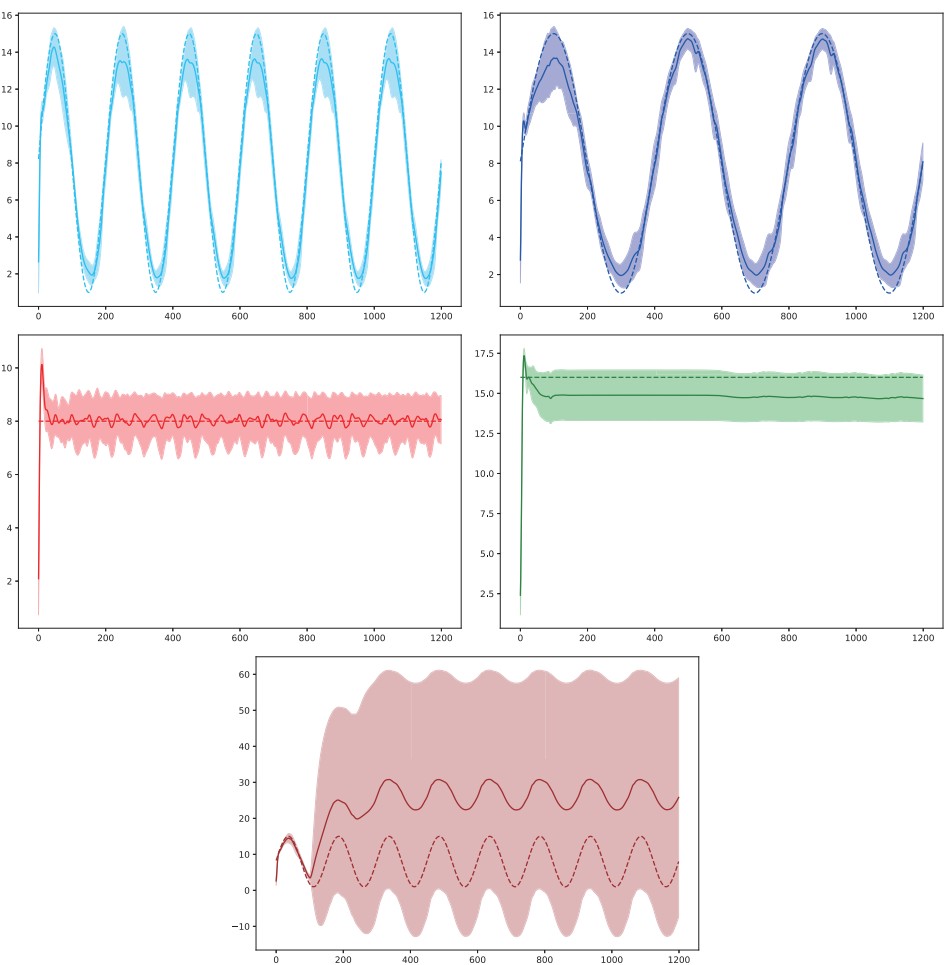

Figure 20: Zoom-in view of Figure 5 (b)

## J Hyperparameters

Table 1: Hyperparameters of LAC

| Hyperparameters | Repressilator | CartPole | FetchReach | HalfCheetah |
|---|---|---|---|---|
| Time horizon $N$ | 5 | 5 | 5 | $\infty$ |
| Minibatch size | 256 | 256 | 256 | 256 |
| Actor learning rate | 1e-4 | 1e-4 | 1e-4 | 1e-4 |
| Critic learning rate | 3e-4 | 3e-4 | 3e-4 | 3e-4 |
| Lyapunov learning rate | 3e-4 | 3e-4 | 3e-4 | 3e-4 |
| Target entropy | -3 | -1 | -5 | -6 |
| Soft replacement($\tau$) | 0.005 | 0.005 | 0.005 | 0.005 |
| Discount($\gamma$) | 0.75 | 1.0 | 1.0 | 0.995 |
| $\alpha_3$ | 1.0 | 1.0 | 1.0 | 1.0 |
| Lyapunov critic network structure | (256,256,16) | (64,64,16) | (64,64,16) | (256,256,16) |

For LAC, there are two networks: the policy network and the Lyapunov critic network. For the policy network, we use a fully-connected MLP with two hidden layers of 256 units, outputting the mean and standard deviations of a Gaussian distribution. As mentioned in section 4, it should be noted that the output of the Lyapunov critic network is a square term, which is always non-negative. More specifically, we use a fully-connected MLP with two hidden layers and one output layer with different units as in Table 1, outputting the feature vector $\phi(s, a)$. The Lyapunov value is obtained by $L_c(s, a) = \phi^T(s, a)\phi(s, a)$. All the hidden layers use Relu activation function and we adopt the same invertible squashing function technique as Haarnoja et al. (2018) to the output layer of the policy network.

