# OpenReview forum: "Model-free Learning Control of Nonlinear Stochastic Systems with Stability Guarantee"
_ICLR.cc/2020/Conference — Reject_

### Official Review · AnonReviewer1 · 2019-10-20
**Official Blind Review #1**

**Rating:** 1

**Review:**



######## Rebuttal Response:

Thanks for the thorough response.

Q2: The title still hasn’t changed on the current draft
Q4: To be more precise:
‘a novel data-based approach for analyzing the stability of the closed-loop system is proposed by constructing a Lyapunov function parameterized by deep neural network’ - this alone is not novel, you would need to specify how your method of doing this is new
‘a practical learning algorithm is designed to search the stability guaranteed controller’ - this is a natural consequence of contribution 1, some further justification is needed as to why this could be viewed as an interesting contribution (i.e. the Lagrangian approach, if this is novel)
‘ the learned controller is able to stabilize the system when interfered by uncertainties such as unseen disturbance and system parameters variations of certain extent’ - this is not a contribution, but an experimental result.

Q5: The review believes that model-free control and stability-guarantees are fundamentally orthogonal ideas, rather than just under-studied work as the authors have been suggesting in the script and rebuttal. Given that discrete-time Lyapunov stability is defined through expressions along the lines of L(f(x)) - L(x) < 0, for Lyapunov function L and closed-loop dynamics f, claiming that stability is being ‘analyzed’ without f is disingenuous. Instead, by making the value function a Lyapunov function, the goal is that the *converged* value function should produce a stable policy, and still, this is surely only assured within the space of samples. Moreover, the use of the discount factor \gamma, popular in MFRL, essentially acts as a time horizon, so I’m not convinced a Lyapunov function learned with a \gamma < 1 can be called stable in the pure infinite-horizon sense.
With this in mind, I think the work would benefit from a revised central claim: that the use of Lyapunov value functions (as an inductive bias) provides more *robust* model-free controllers. I believe this message highlights the value of this work for MFRL, without making false assertions. This, in particular, would highlight the fact that many MFRL algorithms are benchmarked on deterministic environments, and therefore incredible brittle as the experimental results suggest.

Q9: This remark was aimed at earlier in the paper, either the introduction or main section, rather than the experimental section. The fact that a value function can be viewed as a Lyapunov function makes sense but I’m not sure it is a well-known fact in the wider community. Basically, an introduction to the intersection of Lyapunov stability and optimal control would improve the paper.

Q:10 The fact that the clipping of the multiplier corresponds to unstable policies during learning demonstrates that this pitfall needs to be expressed explicitly. Whether the stability guarantees apply to the converged policy or also intermediate policies is not clear on the initial reading of the paper.
For me, this highlights another weakness in the paper. This initial theorems talk of L(s), which relates to the critic L_c by L(s) = E_{u\sim\pi(s)} [L_c(s,u)], however in the subsequent objectives (eg Eq 2), this marginalization never occurs, therefore I don’t feel like you can say Theorem 2 applies to your resultant algorithm.  Moreover, with the \alpha_3 c term in Equation 3, c should be c(s, a) with a marginalized, which it doesn’t appear to be, and the hyperparameter alpha_3 is never discussed nor tuning explained. Assuming this not done in the code, the experiments need to be re-evaluated with Theorem 2 properly enforced through a sample approximation of the marginalization.

Q11/Q12: Thank you for the Markov jump experiments. I’m not sure I understand why LAC is able to learn the task while SAC cannot. To me, this suggests perhaps a lack of hyperparameter tuning for SAC or further investigation. Moreover, there are typos in captions Fig 1, e and f.
A note of figures: Please ensure all axes should be labeled and should be of sufficient size. Many are too small and unreadable. Figure 2 looks like it could be 1 plot (though perhaps requires normalization).
Given that the strength of this method is the added robustness upon convergence, I think it would be valuable to focus less on time-domain results (Figure 3) (these can be added to the appendix for clarity), but instead show how each parameter/noise variation affects the mean and variance of the episodic return. I would expect that, while LAC provides significant robustness, it is still limited. The results don’t demonstrate this. It would also be interesting to know which hyperparameter controls this limit. I imagine there is a robustness/performance tradeoff.

In conclusion, while I appreciate the efforts the authors put into the rebuttal, the extended discussions made me rethink my rating and I have decreased my rating to reject. I believe fixing the issues highlighted above and redrafting the central message must be done before this paper is ready for publication.


######## Review:
This paper investigates the use of Lyanpunov theory as an inductive bias for improving the stability / robustness of policies in a model-free actor-critic reinforcement learning setting. Through viewing the Critic as a Lyapunov function, optimizing the policy with a Lyapunov-based constraint is meant to ensure the stability of the policy through a ‘cost stability’ metric.. Experimental results show that Lyapunov-based Soft-Actor Critic (LAC) is more robust than SAC on some linear and nonlinear environments.

The reviewer believes that the study of intersections between Control Theory and RL to be immensely valuable and the authors outline a principled formulation. However,  the implementation, experiments and general manuscript suggest that paper requires further work before it is conference-ready.

As the author understands it, the current state of the literature of Lyapunov methods for Deep Reinforcement Learning can be summarized as:
Richards et al, 2018, Classify stable region and learn neural Lyapunov function for a safe exploration strategy
Berkenkamp et al, 2018: Classify the stable region via GP, move there for exploration
Chow et al 2018 Constrained MDPs for discrete gridworld environments
Chow et al 2019 Constrained MDPs for continuous environments through a projection on the policy
This work: Actor-Critic constrained policy optimization with a lyapunov-based value function critic

In the introduction and the related work, too much emphasis is put on explaining stability and discussing methods like Model Predictive Control (MPC) which do not benefit the rest of the paper. Additionally, the three contributions listed do not seem particularly novel given the past literature.

The premise of the formulation also presents several unquestioned assumptions and design decisions:
Why model-free RL, as the authors also state that many samples are required to validate stability?
How does the requirement of stability inform the search strategy in this work? Especially as SAC uses a maximum entropy stochastic policy to aid exploration.
Do you really get ‘guarantees’ with sample-based methods? I would expect bounds based on the number of samples
The cost-based measure of stability seems open to abuse - i.e. for the half-cheetah environment only the centre-of-mass horizontal velocity in covered in the cost function, the stability of the embodiment (joint angles and velocities) are ignored. For the Fetch Reacher, a cost function in cartesian space ignores instabilities from kinematic singularities in joint space. I would image the cost function needs to be a measure on the entire dynamic state.

The notion of a Value function as a Lyapunov function is very interesting, and since it was the basis of the work, would have benefitted from more discussion, i.e. for which cost/reward function families the equivalence is valid for, and how it compares to other Lyapunov candidate functions.

With the RL formulation, the requirement of clipping with the lagrangians is suspicious, as it suggests the objective and/or its numerics are not well posed.

With the choice of experiments, they do not seem to question the central problem outlined by the paper. Rather than show environments SAC returns unstable trajectories during learning, the experiments aim to demonstrate instead a general robustness. The reviewer appreciates that stability is difficult to assess; however, while stability is heavily linked to robustness, a paper title promising stability guarantees should demonstrate some strong empirical evidence stability.
Additionally, the choice of environments do not seem to be ideal test beds for stability - i.e, the half-cheetah is stabilized via interactions with the ground. The reviewer would prefer to see simpler nonlinear environments, such as Markov Jump Processes / Switching Linear Dynamics, where SAC clearly demonstrates instability during learning which LAC is sufficiently regularized against. Additionally, while the `repressilator’ is an interesting application to the domain of bioengineering, its addition does not seem to be especially motivated by the central goal of the paper, so just adds to confuse the reader with unnecessary theoretical content.

Moreover, a brief literature review uncovered some relevant earlier work which was not cited:
Construction of neural network based Lyapunov functions, Petridis et al, 2006
Generation of Lyapunov functions by neural networks, Noroozi et al, 2008
Lyapunov Design for Safe Reinforcement Learning, Perkins et al, 2002
Some of the references also appear incorrectly formatted or incorrect, i.e. the reference for Spencer et al, 2018 should be the CoRL 2018 version rather than arxiv.

Also, the general use of grammar in the manuscript would benefit from another draft. In particular, the title could be improved, i.e.
    Model-free Control of Nonlinear Stochastic Systems with Stability Guarantees


**Experience Assessment:**

I have read many papers in this area.

**Review Assessment: Checking Correctness Of Derivations And Theory:**

I carefully checked the derivations and theory.

**Review Assessment: Checking Correctness Of Experiments:**

I carefully checked the experiments.

**Review Assessment: Thoroughness In Paper Reading:**

I read the paper thoroughly.

---

> ### Author Response · Authors · 2019-11-14
> **Thank you for the feedback!**
>
> We would like to thank the reviewer for the evaluation and valuable feedback. We have seriously considered these comments and suggestions. To address these concerns, we carefully revised the manuscript and included more experiments accordingly. The details are explained in the following.
>
> Q1: a brief literature review uncovered some relevant earlier work which was not cited.
>
> A1: We thank the reviewer for pointing out these previous works. We updated the papers to incorporate citations for the suggested citation and other paper on construction/learning of Lyapunov functions.
>
>
> Q2: In particular, the title could be improved, i.e. Model-free Control of Nonlinear Stochastic Systems with Stability Guarantees
>
> A2: We thank the reviewer for the helpful suggestion, the title has now been revised.
>
>
> Q3: In the introduction and the related work, too much emphasis is put on explaining stability and discussing methods like Model Predictive Control (MPC) which do not benefit the rest of the paper.
>
> A3: Thank you for the comment. We have removed the discussion on MPC and reduced the explanation on stability. Considering that the issue of system stability in model-free RL is still an open problem, we believe that the introduction of the concept of stability is still necessary. Thus, the basic explanation and a practical example are still reserved in the revised paper.
>
>
> Q4: Additionally, the three contributions listed do not seem particularly novel given the past literature.
>
> A4: As explained in Q3, a model-free approach for evaluating stability and finding stability assured policies of nonlinear stochastic systems are still missing. To the best of our knowledge, we are not aware of any prior work that could analyze the stability of nonlinear stochastic systems in a model-free manner nor design stability-assured policies solely using samples. In the experiments, we show that the proposed method is superior to state-of-the-art model-free RL algorithms, both in terms of performance, convergence and robustness. In particular, in the example of repressilator, our approach can generalize to different tracking references while others hardly can; and in the example of Markovian jump system where there are several subsystems switching under some condition, our algorithm can yield good performance while others even can hardly converge. We will really appreciate if the reviewer can suggest any prior works that we were possibly missing.
>
>
> Q5: Why model-free RL, as the authors also state that many samples are required to validate stability?
>
> A5: Previous results on guaranteeing stability require a model (either known or learned) of environment and the validation of stability condition on the whole state space (i.e. infinite constraints if the state space is continuous), such as (Meyn'2012; Berkenkamp'2017). Similar to these model-based approaches, we also assume the model is at least a Markov decision process (MDP). However, beyond this, there are always some constraints on the model in the model-based approach, such as Lipchitz continuity, discretization of state space, etc., which limit their applicability. It should be mentioned that for the Markovian jump system where the subsystem is linear like the ones we propose in our revised paper, even the model is completely known, the stability analysis and controller design is notoriously difficult for decades and still very active in control theory community [1,2].
>
> Instead, we show in this paper that it is possible to directly utilize data in analyzing the stability of a stochastic system. However, like many other RL approaches, sample efficiency is an underlying fundamental problem. Without further assumption or knowledge on the system dynamic, the evaluation of stability condition is only valid if enough amount of data is collected.
>
>
> Q6: How does the requirement of stability inform the search strategy in this work? Especially as SAC uses a maximum entropy stochastic policy to aid exploration.
>
> A6: The training procedure is basically the same as SAC. The requirement of stability is passed to the policy through the gradient of Lyapunov critic with respect to the policy. In the meantime, the policy is also guided by the entropy of policy, which is required to sustain beyond a threshold to encourage exploration.
>
> [1] Kwon, Nam Kyu, et al. "Dynamic output-feedback control for singular Markovian jump system: LMI approach." IEEE Transactions on Automatic Control 62.10 (2017): 5396-5400.
> [2] Liu, Ming, Daniel WC Ho, and Yugang Niu. "Stabilization of Markovian jump linear system over networks with random communication delay." Automatica 45.2 (2009): 416-421.

---

> > ### Author Response · Authors · 2019-11-14
> > **Continued, Part 2**
> >
> > Q7: Do you really get ‘guarantees’ with sample-based methods? I would expect bounds based on the number of samples
> >
> > A7: The answer for guarantee is yes. The stability considered in this paper is the general case of mean-square stability and mean partial state stability which is well known in stochastic control theory. If stability conditions are satisfied based on a thorough estimation of state distribution $\mu$ (not the model), then the probability of states converging to zero is one. As shown in the empirical results in Appendix F, eventually all the trajectories converge to equilibrium or track reference signal without exception. As a first step in providing stability guarantee for model-free RL, we show that the stability guarantee could be established in a model-free manner and such guarantee brings promising improvement for the RL agents, such as stable behavior and robustness.
> >
> > As explained in A5, the question on sample bounds is related to the fundamental problem of sample efficiency in RL. We agree that deriving bounds based on the number of samples is extremely important, but is out of the scope of this paper, especially without any further assumption on the system dynamic and completely in a model-free manner. With further assumption on the system dynamic (such as Lipchitz or Gaussian process), it is possible to derive such bound and we leave this for future work.
> >
> >
> > Q8: The cost-based measure of stability seems open to abuse. I would image the cost function needs to be a measure on the entire dynamic state.
> >
> > A8: We appreciate the reviewer for raising this question. Our definition based on cost function is essentially the general case of the classical definition of stability for the stochastic dynamical system in control theory. As discussed in Section 2, we specifically focus on two classes of a cost function which correspond to the classic definition on stability and partial stability in control theory, i.e. norm of state and norm of partial state. We will give further explanations in the following. As the reviewer pointed out, in some cases, the cost function may only capture part of the state. This is formally defined as partial stability in control theory [3]. The partial stability is often necessary and arises from the study of electromagnetics, inertial navigation systems, spacecraft stabilization via gimballed gyroscopes and/or flywheels, combustion systems, vibrations in rotating machinery. For the HalfCheetah example, stability in terms of the full state is hardly useful except proper motion planning algorithm is designed to generate reference signals, while partial stability could be generally adopted in running, walking and path tracking tasks. With the appropriate choice of partial state, e.g., the norm of the difference between speed and desired speed, a task like running at a certain speed could be described by partial stability.
> >
> >
> > Q9: The notion of a Value function as a Lyapunov function is very interesting, and since it was the basis of the work, would have benefited from more discussion, i.e. for which cost/reward function families the equivalence is valid for, and how it compares to other Lyapunov candidate functions.
> >
> > A9: We appreciate the reviewer's valuable suggestion. To address this, we have included additional experiments and discussion on this in Section 5.5. As discussed in Section 2 and the response above, only the norm of state or partial state is taken into consideration.
> >
> > First, we vary the time horizons for the Lyapunov candidates, ranging from 1 to infinity (i.e. value function). For all the horizons, our approach is able to train stabilizing policies. However, the policies possess different levels of robustness: the candidates with the appropriate length of the horizon (N=5) found the most robust policies.
> >
> > Second, we vary the output structure of Lyapunov candidate beyond the quadratic form, such as biquadratic form (power 4) and absolute value form (power 1). It is found that the Lyapunov critics perform comparably well in terms of total cost and robustness. This proves that our framework allows for a general class of Lyapunov critic, as long as the network output is guaranteed to be semi-positive definite.
> >
> > [3] Vorotnikov, Vladimir Ilʹich. Partial stability and control. Springer Science & Business Media, 2012.

---

> > > ### Author Response · Authors · 2019-11-14
> > > **Continued, Part 3**
> > >
> > > Q10: With the RL formulation, the requirement of clipping with the Lagrangians is suspicious, as it suggests the objective and/or its numeric are not well-posed.
> > >
> > > A10: To address the reviewer's concern, we have included figures demonstrating the value of Lagrange multiplier and detailed discussion in Section 5.2 and Appendix G. As shown in figure 2, the clipping of $\lambda$ only takes effect in an early stage of training in a few tasks. In all the tasks $\lambda$ converges well along with the total cost. Clipping the maximum value of $\lambda$ is necessary, in case that $\lambda$ grows too much due to the violation of stability condition in the early stage of training, resulting in the inappropriate step length for the policy update. Clipping is a useful technique to prevent instability of optimization, especially in gradient-based methods, see [4-7]. Conversely, when the stability condition is satisfied, $\lambda$ quickly drops and helps the convergence of the algorithm, which inherently prevents overfitting and enhances robustness and generalization.
> > >
> > > Q11: With the choice of experiments, they do not seem to question the central problem outlined in the paper. More empirical evidence demonstrating stability should be provided.
> > >
> > > A11: We appreciate the reviewer for the helpful suggestions. We have included three more sets of experiments and provided two metrics to evaluate the stability of closed-loop systems. The results are demonstrated in Section 5.1, 5.2, 5.3 and Appendix F. Specifically, we included two simple but challenging Markovian jump systems for which other algorithms can even hardly converge and a complicated version of Repressilator to evaluate LAC and baseline methods.
> > >
> > > As suggested by Reviewer 2 in Q2, we also included a new baseline method which exploits the Lyapunov method in model-free RL (chow'19).
> > >
> > > The evaluation of stability is in two folds. First, we evaluate the trained policies in the original training environment and observe the state trajectories of the closed-loop system. Each trained policy is tested for numerous trials in the environment and all the paths are shown in Figure 11 and 12. As shown in the figures, without stability guarantee, the state trajectories either diverge (see Figure 11b and Figure 12d), or continuously oscillate around the reference trajectory or equilibrium (see Figure 11d and Figure 12b). In the MJS examples, the trajectories even diverge to or oscillate in unacceptable magnitude (1e7 and 1e10). Contrarily, the stability assured method stabilizes the system well in all tasks (i.e. the state trajectories converge to the reference signal or equilibrium).
> > >
> > > Secondly, stability guarantee could be validated by checking the value of corresponding Lagrange multiplier. LAC aims to solve the dual problem of the original policy optimization problem, where the Lagrange multipliers are updated by gradient ascent. When (2) is satisfied, $\lambda$ will continuously decrease until it becomes zero. More specifically, the decline of $\lambda$ implies the satisfaction of stability condition at that update; $\lambda$ only converges to zero if the stability condition is assured. As shown in Figure 2, across all the experiments, $\lambda$ converges stably to zero and maintain zero thereafter, which implies the satisfaction of stability condition in all the training trials.
> > >
> > > Q12: The reviewer would prefer to see simpler nonlinear environments, such as Markovian Jump Processes/Switching Linear Dynamics, where SAC clearly demonstrates instability during learning which LAC is sufficiently regularized against.
> > >
> > > A12: We would like to thank the reviewer for the valuable suggestions. We have included two Markovian jump systems (MJS) and a repressilator system with unstable oscillation behavior to provide a more insightful evaluation of our stability assured method. Surprisingly, the nonlinear environment is simple, but the control is very difficult using the state-of-the-art RL algorithms. It is perhaps due to the random and abrupt changing in system dynamics. Moreover, the adopted examples contain subsystems that are unstable and even uncontrollable, which make the tasks even harder. Summary on the empirical results are referred to A9, Section 5.2 and Appendix F.
> > >
> > > We believe that these examples have revealed a meaningful future direction: developing switching Lyapunov-based model-free control methods to further improve the control performance of such hybrid systems.
> > >
> > > [4] Schulman, John, et al. "Proximal policy optimization algorithms." arXiv:1707.06347 (2017).
> > > [5] Wang, Ziyu, et al. "Dueling network architectures for deep reinforcement learning." arXiv:1511.06581 (2015).
> > > [6] Bello, Irwan, et al. "Neural combinatorial optimization with reinforcement learning." arXiv:1611.09940 (2016).
> > > [7] Bengio, Yoshua, et al. "Advances in optimizing recurrent networks." 2013 IEEE International Conference on Acoustics, Speech and Signal Processing. IEEE, 2013.

---

> > > > ### Author Response · Authors · 2019-11-14
> > > > **Final part**
> > > >
> > > > Q13: Additionally, while the repressilator is an interesting application to the domain of Bioengineering, its addition does not seem to be especially motivated by the central goal of the paper, so just adds to confuse the reader with unnecessary theoretical content.
> > > >
> > > > A13: Control of repressilator is a very important topic in control engineering, see work of Richard Murray, Mustafa Khammash, Domitilla del Vecchio, Antonis Papachristodoulou, Guy-Bart Stan who applied the concept and tools in control engineering for synthetic biology. First of all, we try to introduce a minimal biology background as possible without confusing the readers in the revised version. Second, similar to the examples of the CartPole, HalfCheetah and FetchReach, for the audience who do not have the background of Robotics, one can simply focus on the mathematical description of the system, i.e., the nonlinear ordinary differential/difference equations for repressilator in Equation D.1 and D.2 in the Appendix. Third, the very reason we select this example is not only its importance in Bioengineering but also its interesting oscillation behaviour and inherent difficulties to be controlled [8], compared to the examples of CartPole, HalfCheetah and FetchReach.
> > > >
> > > > In the experiment, we found that SPPO and SAC could not even converge (see figure 1). In terms of online inference, the repressilator system controlled by SAC agents either diverge or oscillate continuously (see figure 11). On the contrary, the LAC agents successfully stabilize the system in all the test trials.
> > > >
> > > > [8] Strogatz, S. H. (2018). Nonlinear dynamics and chaos: with applications to physics, biology, chemistry, and engineering. 2018, CRC Press.
> > > >
> > > >
> > > > Lastly, if there is any misunderstanding of the reviewer's comment, or the reviewer has any further concerns, please do let us know.

---

### Official Review · AnonReviewer2 · 2019-10-24
**Official Blind Review #2**

**Rating:** 3

**Review:**

In this work the authors studied the model-free RL approach for learning a policy with stability guarantees. Leveraging the Lyapunov stochastic stability criterion, instead if minimizing the cumulative cost (plus a soft entropy), they propose optimizing an objective function with a specific Lyapunov critic, which is a specific critic function that satisfies the Lyapunov criterion to guarantee stability. They also show in several Cartpole, Mujoco, and Repressilator experiments that this approach is more robust to perturbations (such as sinusoids), where the agent are more robust to dynamic uncertainties and disturbances.

In general, the topic of guaranteeing stability is a topic in safe RL, and I find this work of enforcing stability in model-free RL interesting. Through the specific parameterization of quadratic Lyapunov function (in the latent space), the authors proposed learning a new critic function that is a value function but at the same time (almost) satisfies the Lyapunov constraints. While this is an interesting idea, and the experimental results look promising, I do have several questions.

First, regarding the learning problem of Lyapunov function, how does the proposed way of learning L differ from the one in Richard'18: The lyapunov neural network: Adaptive stability certification for safe learning of dynamic systems, where the problem is formulated as a classification (while in here it is a regression problem)?
Second, while this approach is intuitive, since the approach is penalty-based (Lagrangian based), I do not see how the Lyapunov criteria in Theorem 1 is guaranteed, in this case is stability guaranteed by the policy learning  algorithm? If not, what do the authors do to enforce that?
Third, if one formulates the immediate constraint cost of the CMDP to be the distance of the state to the equilibrium point,  then the (undiscounted, shortest-path type)  CMDP total cost constraint should guarantee stability (because the total distance cumulative cost is bounded, meaning that the distance cost converges to zero). Then, one can use the Lyapunov approach by Chow'19 (in modulo to their setting in discounted MDPs) to enforce stability (which is a specific notion of safety in this case).  How does the proposed method compare with this approach? Can the authors provide numerical comparisons with the method proposed by Chow'19 as well?

**Experience Assessment:**

I have published one or two papers in this area.

**Review Assessment: Checking Correctness Of Derivations And Theory:**

I assessed the sensibility of the derivations and theory.

**Review Assessment: Checking Correctness Of Experiments:**

I assessed the sensibility of the experiments.

**Review Assessment: Thoroughness In Paper Reading:**

I read the paper at least twice and used my best judgement in assessing the paper.

---

> ### Author Response · Authors · 2019-11-14
> **Thank you for the feedback!**
>
> We would like to thank the reviewer for valuable feedback. We have seriously considered these comments and suggestions and carefully revised the manuscript accordingly. The details are explained below
>
>
> Q1: How does the proposed way of learning L differ from the one in Richard'18?
>
> A1: First of all, it should be claimed that the approach in Richard'18 is a model-based approach, where a mathematical model is used, by exploiting the approach in Berkenkamp'17. The Lyapunov neural network updates together with policy to maximize the region of attraction (ROA). However, the Lyapunov decrease condition is checked on a discretization of state-space point-wisely, which limits its use in the tasks with low-dimensional limited state space.
>
> Instead, our approach developed a sample-based model-free stability theorem that does not require such discretization (only one stability condition to be validated) and is applicable to high-dimensional control tasks, such as Half-cheetah. Our approach proposes to select a certain Lyapunov candidate and uses a non-negative network to approximate it. Thus many choices of network parameterization are applicable, such as the absolute value $L(s)=\sum_{j=1}^m|\phi(s)|$ or fourth power $L(s)=\sum_{j=1}^m \phi^4_j(s)$, where $\phi(s)$ is a neural network with $m$ dimensional output. To show this, we have included additional experiments in Section 5.4 and compare these different selections of Lyapunov candidate in terms of performance and robustness. As shown in figure 5 (c,d), LAC converges well with different Lyapunov parameterization and they possess similar robustness to impulsive forces. Therefore, the selection of Lyapunov candidate with different forms does not affect the performance much.
>
>
> Q2: How is the stability criteria guaranteed by the algorithm?
>
> A2: We appreciate the reviewer for this helpful comment. LAC aims to solve the dual problem of the original policy optimization problem When stability condition (2) is satisfied, $\lambda$ will continuously decrease until it becomes zero. Thus, by checking the value and variation of $\lambda$, the satisfaction of stability condition during training and at convergence could be validated. More specifically, the decline of $\lambda$ implies the satisfaction of stability condition at that update; $\lambda$ only converges to zero if the stability condition is assured. We have modified the manuscript and shown more empirical results to prove this, see Section 5.3 and Appendix G.
>
> This question is also related to Q8 of Reviewer 1 on the issue of clipping with the Lagrangians. The Reviewer 2 can also see A8 for Q8 for further explanations.
>
>
> Q3: How does the proposed method compare with Chow'19? Can the authors provide numerical comparisons with the method proposed by Chow'19 as well?
>
> A3: We appreciate the reviewer’s valuable suggestion. In the revised paper, we have included more experiments to compare the algorithm (SPPO) proposed by Chow’19 with LAC, see Section 5.2, 5.3 and Appendix F. As shown in figure 1, SPPO could only perform comparably with LAC in the Cartpole and FetchReach examples. In all the other examples, SPPO fails to find a stabilizing policy while LAC succeeds without exception. In terms of robustness, SPPO agents appear to be vulnerable to both parametric uncertainty and perturbations and could not achieve the same performance as LAC.
>
> SPPO was originally developed to deal with safety-related tasks and has a strength of guaranteeing safety even during training. However, although the Lyapunov method is exploited, the approach is not aimed at providing stability guarantee. In their framework, the discounted sum of cost is chosen to be the Lyapunov function. This discounting in cost is a fundamental setting this framework, which further facilitates the construction of auxiliary constraint cost, and deriving the policy improvement algorithm. Thus, introducing an un-discounted, shortest-path sum of the cost to their framework requires thorough and completely different derivations. In the current framework, SPPO could only guarantee the discounted sum of the cost to be kept under a certain threshold.
>
> Lastly, if there is any misunderstanding of the reviewer's comment, or the reviewer has any further concerns, please do let us know.

---

### Official Review · AnonReviewer3 · 2019-11-01
**Official Blind Review #3**

**Rating:** 6

**Review:**

In this paper, the authors introduce an algorithm to learn a stable controller using deep NN actor-critic method. They define the stability in the mean cost criteria,  which is used to constrain the critic network as a Lyapunov function. In addition, the semi-positive definiteness of the Lyapunov function is enforced by constructing the critic.
The problem is important to control with deep RL. The paper is written clearly. The reviewer has the following questions regarding the stability of the learned policy.

- How is the stability in the mean cost related to the stability of stochastic systems? See, for example, the Lyapunov stability of stochastic systems (survey in [1])?
- The authors enforce semi-positive definiteness using the construction of the value function approximation as the quadratic function bases. Then the semi-negative definiteness is enforced using penalty on the Lyapunov stability of the critic. Then the target network is trained to minimize the difference between the target and the critic. The question is, how is the stability of the target ensured by minimizing the difference with a Lyapunov critic? Is it possible to have an unstable target function that happens to minimize the distance?

[1] H. J. Kushner, “A partial history of the early development of continuous-time nonlinear stochastic systems theory,” Automatica, vol. 50, no. 2, pp. 303–334, 2014.

**Experience Assessment:**

I have read many papers in this area.

**Review Assessment: Checking Correctness Of Derivations And Theory:**

I assessed the sensibility of the derivations and theory.

**Review Assessment: Checking Correctness Of Experiments:**

I assessed the sensibility of the experiments.

**Review Assessment: Thoroughness In Paper Reading:**

I read the paper at least twice and used my best judgement in assessing the paper.

---

> ### Author Response · Authors · 2019-11-14
> **Thank you for the feedback!**
>
> We would like to thank the reviewer for valuable feedback. We have seriously considered these comments and suggestions and carefully revised the manuscript accordingly. The details are explained below
>
>
> Q1: How is the stability in the mean cost related to the stability of stochastic systems?
>
> A1: In this paper, two kinds of cost functions are considered, i.e., the norm of full state and the norm of partial state tracking error.
>
> First, when the norm of the full state is chosen to be the cost, the stability in mean cost is equivalent to mean square stability [1], i.e. the state converges to zero as time goes to infinity. The stability discussed in [2] is called almost-sure stability, which is also a notable definition of stability in stochastic systems and implies that almost all the sample paths of the system state will converge to zero. In short, mean square stability is stricter than almost sure stability and under some conditions, they are equivalent (see Theorem 2 in [3]).
>
> Second, when the norm of partial state tracking error is chosen to be the cost, then the stability in mean cost is equivalent to the partial state stability, which requires partial dimensions of the state converging to zero. Mean square stability is a special case of partial state stability, in which partial state is the full state and reference signal is 0. Overall, the stability in mean cost is a more general definition of stability and the classical definitions could be viewed as special cases of it.
>
>
> Q2: How is the stability of the target ensured by minimizing the difference with a Lyapunov critic? Is it possible to have an unstable target function that happens to minimize the distance?
>
> A2: To address the reviewer’s question, we have included more empirical results to show the stability of both the closed-loop system and the convergence (or stability) of the learning process, see Section 5.2. As shown in the empirical results, the LAC agents stabilize the systems well in all tasks. All the state trajectories converge to the reference signal or equilibrium eventually. In terms of the training process, LAC converges stably in all the experiments. The convergence (or stability) of $L_c$ and policy learning is not affected by the stability of the stochastic system. Even though the original system is unstable, the algorithm is still able to learn a stabilizing policy and corresponding $L_c$ (see the Repressilator and Markovian jump system examples).
>
> The learning process of the proposed algorithm is composed of two parts, the learning of Lyapunov critic $L_c$ and learning of policy. $L_c$ is updated to minimize the difference between the chosen Lyapunov candidate $L_{target}$ (such as the finite-horizon sum of cost and the value function). The policy is updated to satisfy the stability condition in (2) that is formulated by $L_c$, so that the stability of the stochastic system is ensured (in mean cost).
>
> When $L_{target}$ is chosen to be the value function, we introduce a “target network” which is updated by soft replacement to help stabilize the training, as in [4]. When $L_{target}$ is chosen to be the finite-horizon sum of cost as in (6), the training of $L_c$ becomes a supervised learning problem. The supervised learning is more stable than the value function learning since explicit approximation target is provided. Through an experiment in Section 5.4, we further show that both these candidates converge stably, though they do differ in terms of robustness of learned policies.
>
> [1] Boukas, El-Kebir, and Z. K. Liu. "Robust $H_\infty$ control of discrete-time Markovian jump linear systems with mode-dependent time-delays." IEEE Transactions on Automatic Control 46.12 (2001): 1918-1924.
> [2] H. J. Kushner, “A partial history of the early development of continuous-time nonlinear stochastic systems theory,” Automatica, vol. 50, no. 2, pp. 303–334, 2014.
> [3] Kozin, Frank. "A survey of stability of stochastic systems." Automatica 5.1 (1969): 95-112.
> [4] Mnih, V., Kavukcuoglu, K., Silver, D., Rusu, A. A., Veness, J., Bellemare, M. G., Graves, A., Riedmiller, M., Fidjeland, A. K., Ostrovski, G., et al. Human-level control through deep reinforcement learning. Nature, 518(7540):529–533, 2015.
>
>
> Lastly, if there is any misunderstanding of the reviewer's comment, or the reviewer has any further concerns, please do let us know.

---

### Author Response · Authors · 2019-11-14
**For all reviewers, paper updated to address the feedback**

We have addressed all the reviewer's concerns and revised the manuscript. The changes are marked in blue. We would appreciate if the reviewers could take a look at our changes and let us know if they would like to revise their rating or request additional changes that would alleviate their concerns.

The new reproducible code is in the same Dropbox folder.

In summary, here are the main changes that we made to the paper:

- Extended the related work Section to incorporate citations for Lyapunov function learning/construction.

- Deleted unnecessary explanation on stability in the introduction.

- Included more environments as testbeds for stability, including two Markovian jump systems and a complicated Repressilator system. Details of the experiments are in Section 5.1, 5.2 and Appendix D and E.

- Included an additional baseline method, SPPO (a Lyapunov-based RL method) based on Chow’19, and comparison is made. See Section 5.2 and Appendix H.

- Included more evidence and experiments on stability guarantee, where LAC (our method) demonstrates stable behaviour while the baseline method fails. See Section 5.2 and Appendix F.

- Included more evidence on the convergence of LAC and satisfaction of stability condition of trained policies. See Section 5.3 and Appendix G.

- Edited Section 5.5 to compare the performance of LAC with different Lyapunov critic structures.

---

### Decision · Program_Chairs · 2019-12-19

**Decision:**

Reject

**Comment:**

The authors propose a method to guarantee the stability of a learnt continuous controller by optimizing the objective through a Lyapunov critic. The method is demonstrated on low dimensional continuous control problems such as cart pole.

The reviewers were mixed in their opinion of the paper, especially after the authors' rebuttal. The concerns center around some of the authors' claims regarding theoretical results, in particular that stability guarantees can be asserted for a model-free controller. This claim seems to be incorrect especially on novel data where stability cannot be guaranteed, thus indicating that 'robust controller' might be a better description. There are also concerns about the novelty and the contributions of the paper. Overall, the method is promising but the claims need to be carefully written. The recommendation is to reject the paper at this time.